# Adaptive and context-aware volumetric printing

Sammy Florczak[1,2], Gabriel Größbacher[1], Davide Ribezzi[1], Alessia Longoni[1], Marième Gueye[1], Estée Grandidier[1], Jos Malda[1,2] & Riccardo Levato[1,2 ✉]

We introduce Generative, Adaptive, Context-Aware 3D Printing (GRACE), a new approach combining 3D imaging, computer vision and parametric modelling to create tailored, context-aware geometries using volumetric additive manufacturing. GRACE rapidly and automatically generates complex structures capable of conforming directly around features ranging from cellular to macroscopic scales with minimal user intervention. Here we demonstrate its versatility in applications ranging from synthetic objects to biofabrication, including adaptive vascular-like geometries around cell-laden bioinks, resulting in improved functionality. GRACE also enables precise alignment of sequential prints, as well as the detection and overprinting of opaque surfaces through shadow correction. Compatible with various printing modalities[1–4], GRACE transcends traditional additive manufacturing limitations in automating overprinting and adapting the printed designs to the content of the printable material. This opens new possibilities in tissue engineering and regenerative medicine.

The additive manufacturing workflow, originally conceived more than 40 years ago, has largely remained unchanged. Although 3D printing plays a key role in medical devices, microfluidics, bioengineered tissues, as well as driving innovation in the automotive and aerospace sectors[5–7], the printing process always begins with users defining the desired part by means of computer-aided design (CAD) software, which is then translated to the printer and fabricated either layer by layer or using volumetric methods[1,2,8]. Further research on embedded sensors and feedback loops aims to improve automation and is making substantial strides towards performing in-line quality control of printed objects[9–11]. Yet, 3D printers remain primarily tools that passively execute a command while being agnostic to the composition and nature of the environment in which the printing process takes place.

Enabling printers to detect and respond to contextual cues can open new avenues for many applications, including soft robotics, hierarchical composites and the bioprinting of living cells and human tissues. In fact, the functionality of such systems is intimately linked to both their architecture and relative positioning of their individual components (that is, particles, fibres, living cells)[12,13], whose precise patterning within printed objects cannot be fully controlled. Recent advances in computer vision and artificial intelligence have the potential to greatly enhance this approach. Concurrently, the advent of volumetric printing, including tomographic volumetric additive manufacturing, has enabled the extremely fast production of large parts with virtually unconstrained design freedom, using visible light fields to polymerize photo-responsive resins in a layerless fashion. Owing to its contactless nature, volumetric printing excels at overprinting—that being, non-invasively printing onto or across existing objects, even when produced with other techniques[14]. This includes building multimaterial structures[15] and safely encapsulating fragile living cells and organoids[16].

Such attributes position volumetric printing as an ideal demonstration platform for new fabrication pipelines that would be challenging to implement using traditional technologies.

In this study, we report a new technique to equip 3D printers with the ability to map the composition (chemical and architectural) of the printable material and to take autonomous, informed decisions on which geometries to print. This new workflow allows the printer to produce, within seconds, volumetric, generative designs that adapt and conform to embedded features within the resins, enabling exploration of diverse applications in data-driven additive manufacturing. It also supports complex materials, including living cells with biologically friendly hydrogels, and automates overprinting to create complex multicomponent structures, including models comprising several tissue types (that is, vascularized tissues; bone and cartilage in osteo-chondral models), and mechanical joints with interlocked movable parts, as examples among a broad array of possible printable designs.

## 3D printing guided by volumetric imaging

We termed this workflow GRACE, short for Generative, Adaptive, Context-Aware 3D Printing. We first demonstrated GRACE by integrating volumetric printing with light-sheet imaging. The light sheet rapidly maps the printing volume in 3D to extract positional, morphometric and spectral (that is, fluorescence) information from the contents of the vial. This serves as input for multiparametric modelling algorithms that process the data to generate precisely targeted geometries for printing, effectively enabling adaptive fabrication. Although GRACE printing is relevant for various fabrication processes, we first showcased its key capabilities in the context of bioprinting, using cell-laden hydrogels as resins. In living organs, key tissue components, including structures

[1]Department of Orthopaedics, University Medical Centre Utrecht, Utrecht University, Utrecht, The Netherlands. [2]Department of Clinical Sciences, Faculty of Veterinary Medicine, Utrecht University, Utrecht, The Netherlands. ✉e-mail: r.levato@uu.nl

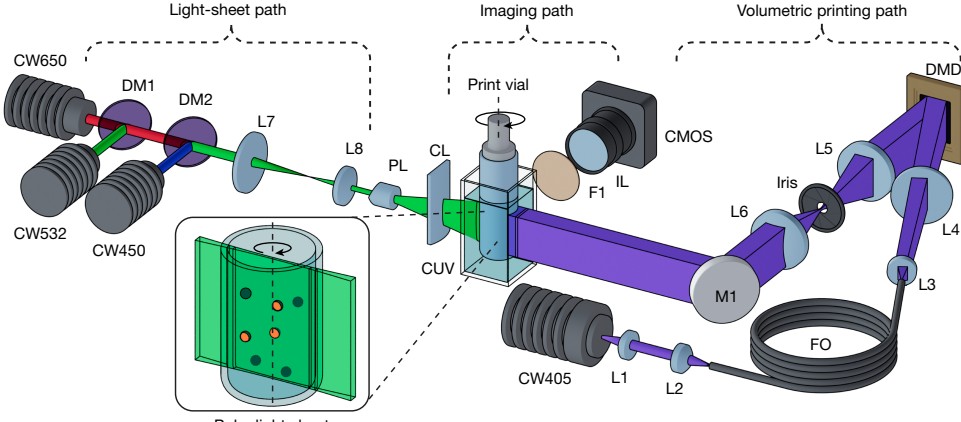

**Fig. 1 | Experimental GRACE printing.** Schematic of the experimental device, with the light sheet (green), imaging and printing (violet) optical paths indicated. The printing path consists of a 405-nm continuous-wave laser source (CW405), collimating lens (L1), fibre coupling lens (L2), square core multimode-fibre optic (FO), collimating lens pair (L3 and L4), DMD, a 1:1 magnification 4*f* relay and Fourier filter (L5, Iris and L6), and a folding mirror (M1). The light-sheet path comprises three collimated laser sources at 450 nm, 532 nm and 650 nm (CW450, CW532 and CW650, respectively), combined by means of dichroic beam-combining optics (DM1 and DM2), 2.5× beam-reduction optics (L7 and L8), a 30° Powell lens (PL) and cylindrical lens (CL) for focusing the light sheet. Along the imaging path, a monochromatic CMOS sensor with imaging lens (IL) captures fluorescent signals by means of a fluorescence imaging filter (F1). All three optical paths intersect on the central axis of a resin-filled print vial, coupled to a rotational stage. To minimize refractive errors during both printing and imaging, the vial itself is immersed within a liquid-filled (refractive-index-matching fluid, depending on the resin to be printed, but typically water when working with low-concentration hydrogels) quartz cuvette (CUV) whose optical faces are precisely aligned orthogonal to each path.

composed of several cells, develop to adapt to the needs of the resident cells. For instance, blood vessels grow into intricate networks to reach each individual cell, ensuring access to nutrients. At present, printing technologies cannot fully recapitulate this process, as cells are randomly or homogeneously distributed within a printed hydrogel. With GRACE, we demonstrated on-the-fly generation of 3D models to create positive and negative features, including targeted vessel-like channel networks that can precisely reach cells, cell clusters and organoids of interest, resulting in improved functionality of the bioprinted cells. We further demonstrated the production of interconnected geometries and the precise encapsulation of various features embedded within the resin. Moreover, we enabled the automated alignment of new prints onto pre-existing ones, permitting the generation of multicomponent constructs. GRACE also enabled the mapping of opaque features, countering shadowing artefacts and improving (over)printing quality.

The experimental device for GRACE comprises two main components: a custom-made tomographic volumetric printer and a light-sheet microscopy path (Fig. 1). We chose polar light-sheet microscopy as the primary scanning modality for its capacity for rapid, large-scale imaging and its ease of incorporation within our set-up, requiring no modification to the printing path. Light-sheet generation could be achieved through two methods: (1) by encoding a light-sheet pattern (that is, a single column of activated pixels) onto the digital micromirror device (DMD) or (2) using an external laser source with dedicated optics. Although the former required no extra hardware, for most of our experiments, we opted for an external light-sheet configuration, maximizing power and signal-to-noise ratio during scanning.

To enable the 3D mapping of features embedded within the printing volume, we developed a protocol that synchronized the motion hardware with the imaging and printing systems (Extended Data Fig. 1a, Supplementary Video 1 and Supplementary Methods 1–4). On initial homing of the vial, the light sheet acquires images that serve as the input of the computer-vision routines. The resultant fluorescent emission of the optical section is captured by the camera at many angular increments to obtain a dense set of polar cross-sections through the axis of the vial. The process can be repeated for different fluorescence channels. The extracted data are then assigned corresponding per-pixel Cartesian coordinates, producing a 3D point cloud in print-volume coordinate space. Depending on the scanned features and intended

model, the raw data can either be sent directly to parametric modelling software or undergo further processing in the form of cluster detection. For this, we used density-based spatial clustering of applications with noise (DBSCAN), chosen for its efficiency with large datasets, ability to identify arbitrarily shaped clusters, automatic outlier exclusion and lack of requirement for pre-defining the cluster count[17]. Clustering detection was applied when numerical indexing of individual features was required or when centroid coordinates were preferred, as is the case with symmetrical features such as particles, organoids or microspheres. We tested our imaging and feature-detection accuracy with suspensions of standardized polyethylene microspheres (diameter 150 or 500 μm), acquiring images using both a DMD-generated and an external light sheet. The results were benchmarked against conventional microscopy, showing no difference in size detection accuracy (Supplementary Methods 5 and Supplementary Fig. 1). This workflow provided robust feature detection in complex, heterogeneous samples, with the resultant coordinate data passed to the parametric modelling software to automatically generate geometries targeted around those scanned features (Extended Data Fig. 1b and Supplementary Figs. 2–4).

## Context-driven parametric structures using GRACE

As a testing platform for printing context-driven architectures, we produced fluorescently stained alginate microspheres of various radii (0.15–0.90 mm, a size range compatible with that of organoids[18]) (Supplementary Methods 6). These were mixed into a gelatin methacryloyl resin (GelMA, 10% w/v), with lithium phenyl-2,4,6-trimethylbenzoylphosphinate (LAP) as a photoinitiator. Using only spatial information with clustering detection to determine the centroids of the alginate particles, we first successfully demonstrated context-aware prints with vascular-like channel networks (diameter 450 ± 20 μm) targeted around individual spheres, with a 300-μm offset (Fig. 2a, Supplementary Methods 7 and 8, Supplementary Figs. 5–7 and Supplementary Video 2). Further proof of perfusability of complex vascular-like structures with different hydrogel resins was also shown (Supplementary Methods 9 and Supplementary Fig. 8). We also fabricated interconnections between several particles (Fig. 2b) and showed the encapsulation of individual features (Fig. 2c). Furthermore, discriminating different particle subpopulations is possible through processing spectral and

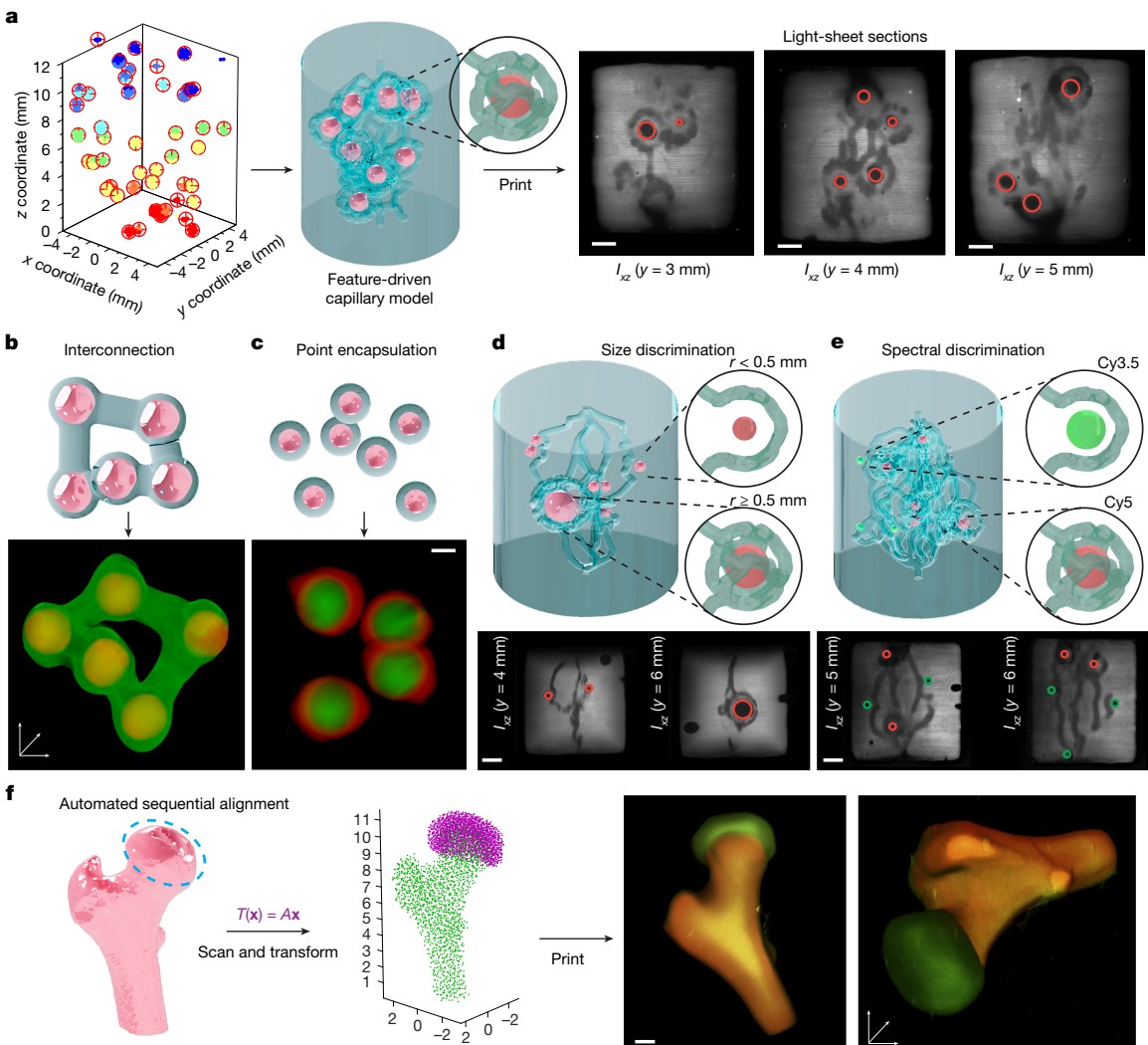

**Fig. 2 | GRACE allows printing adaptive and feature-driven prints with complex geometries.** A showcase of GRACE printing by generating targeted features around randomly distributed, fluorescently stained alginate spheres detected within the resin. **a**, A spherically wrapped channel network architecture is generated around the alginate spheres, showing the process from acquiring the raw data to generating the target model (shown rendered) and, finally, printing and imaging the resultant construct. The light-sheet sections at different depths of the printed gel show in greyscale the stained GelMA and the cross-sections of the convoluted channels printed within the gel, whereas the unstained particles are circled in red (or in green) to improve visualization and to distinguish them from cross-sections of the channels. Scale bars, 1 mm. **b**, Interconnection of randomly distributed alginate spheres with printed struts, showing a render of the target geometry (blue) and resultant light-sheet reconstruction in 3D after printing. **c**, Encapsulation, with corresponding light-sheet 3D reconstruction post-printing indicating presence of spheres. Scale bar, 1 mm. **d**,**e**, Parametric discrimination of generated features based on size (**d**) or spectral emission (**e**). Here different populations receive either a single grazing channel or a more complex spherical channel network. Scale bars, 2 mm. **f**, Automated alignment of a cartilage model to femoral head in a two-part sequential print, with computation for the alignment (post-scanning) taking <5 s to perform on a common personal computer. Our transformed cartilage model was printed directly onto the femoral head, resulting in a multicomponent sequentially printed construct with correct relative positioning of its components. Scale bar, 2 mm.

morphometric data. Polydispersed alginate particles were detected and their individual radii (0.15–0.90 mm) measured by means of cluster detection. Using this information, GRACE generated conditional geometries defined by a size threshold: spheres with $r < 0.5$ mm received a single grazing channel, whereas those with $r \geq 0.5$ mm were enclosed by a more complex spherically wrapped channel network (Fig. 2d). Moreover, we applied a similar conditional approach to discriminate between different fluorophores. Here the model generated solitary grazing channels around Cy3.5-stained microparticles, whereas spherically wrapped networks were generated around Cy5-stained particles (Fig. 2e). Scanning, feature isolation, model generation and printing were completed within approximately 4 min (per spectral channel), underscoring the compatibility with the rapid, high-throughput capabilities of volumetric printing. Detailed processing times are reported in Extended Data Table 1.

Finally, we demonstrated multistep fabrication through automatic alignment (Fig. 2f), eliminating the need for manual repositioning of sequential prints—a process[1,19] that is error-prone and prohibitively slow. GRACE automatically detects and aligns with previously printed structures, enabling the precise positioning of subsequent prints relative to existing ones, streamlining the fabrication workflow to build complex, multilayered or hierarchical structures with high repeatability. We successfully demonstrated this by first printing a model of a human femur, washing it and resuspending it in GelMA at a random orientation. Through an iterative closest point algorithm, the optimal rigid transformation that aligns a pre-existing reference model (correctly positioned cartilage layer) to the scanned point cloud of the femur was identified (Fig. 2f). Our experiments demonstrated how GRACE adapts to various embedded objects (Extended Data Fig. 2)

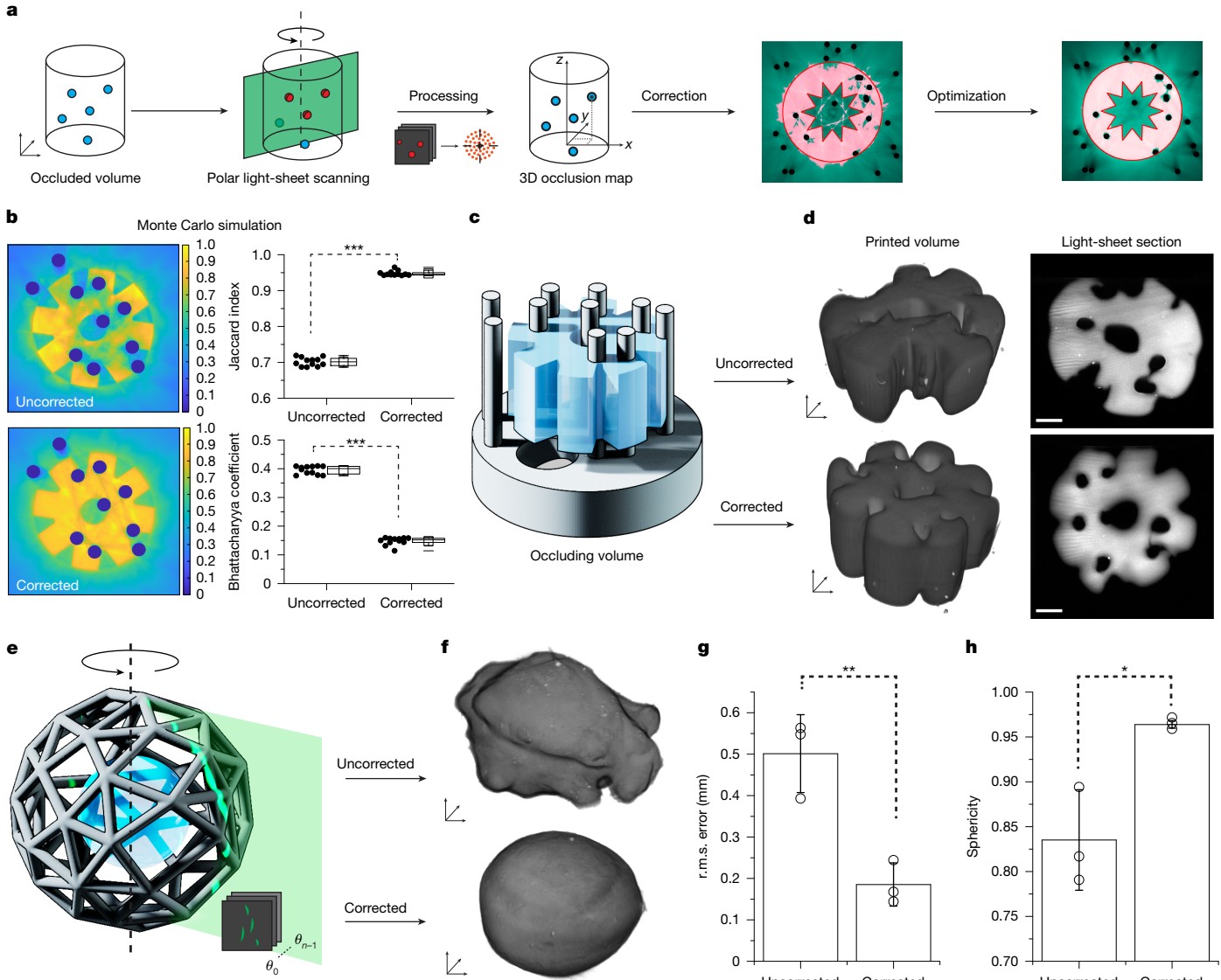

**Fig. 3 | Light-sheet mapping of occluding structures and shadow correction.**
**a**, Flow chart demonstrating the process of scanning, mapping and correcting for the presence of occlusions. **b**, OSMO-based reconstruction of a cog-like target model without and with shadow correction when influenced by ten randomly distributed pillar-like occlusions. Jaccard index and Bhattacharya coefficient demonstrate the relative improvements attained by the correction (mean ± s.d., *t*-test, *n* = 12, ***P < 0.001, DF = 22). **c**, Rendering showing the SLA-printed pillar occlusions and cog-like target geometry to be volumetrically printed around them. **d**, 3D light-sheet reconstruction of corrected and uncorrected prints, as well as a single optical section of the cross-section of each. Scale bars, 2 mm. **e**, Render showing the ball-in-cage model, with spherical target geometry inside. **f**, Light-sheet reconstruction of the resultant prints following the destructive removal of the occluding cage following printing. **g**, r.m.s. error of the printed part. A lower value indicates less deviation from the target geometry (mean ± s.d., *t*-test, *n* = 3, *P* = 0.0071, DF = 4, **P < 0.01). **h**, Sphericity of each printed sample. A higher value indicates that the sample is more spherical (mean ± s.d., *t*-test, *n* = 3, *P* = 0.0137, DF = 4, *P < 0.05).

and generates precise, functional geometries with minimal user input after the initial parametric model definition. This level of automation and adaptability would be impractical, if not impossible, to achieve through manual positioning and 3D modelling.

## Image-guided printing across opaque features

Shadowing artefacts from light-absorbing features is a common challenge in light-based printing. Occlusions within the projection path result in poor reconstruction quality, compromising dimensional accuracy and printing homogeneity when performing overprinting, especially for volumetric printing[1,20]. Addressing this challenge, we applied our light sheet as a profilometer, using the reflected signal to map the surfaces of occluding structures (Fig. 3a). To mitigate printing

artefacts, we used object-space model optimization (OSMO) of tomographic reconstructions to iteratively optimize our tomograms for the presence of these occlusions[21,22]. This was demonstrated in two ways. First, we used an opaque, polymeric occlusion, consisting of ten 1-mm-diameter vertical pillars to provide a reproducible occlusive feature within our build volume. The quantity and spatial distribution of pillars was based on a Monte Carlo optimization by minimizing the Bhattacharyya coefficient (for contrast) and maximizing the Jaccard similarity index—both derived from the OSMO-corrected reconstructions (Fig. 3b and Supplementary Fig. 9). The resulting occlusion phantom (Fig. 3c) was embedded into our resin and scanned, to parametrically generate the representative occlusive geometry based on the pillar centroids and angles (Supplementary Methods 10 and 11 and Supplementary Fig. 10). On optimizing and printing the target geometry

(eight-toothed cog model; Fig. 3b), the shadow-corrected structure demonstrated superior print quality compared with uncorrected output, with the latter showing regions of both under-crosslinking and over-crosslinking within the same print (Fig. 3d). This indicated that there would exist no possible dosage condition for which an accurate output could be attained without shadow correction. Meanwhile, the corrected output showed a more homogenous crosslinking behaviour, with finer details retained across the entire construct. This is corroborated by the Monte Carlo simulation, indicating improvements in both contrast—with the Bhattacharyya coefficient decreasing from $0.39 \pm 0.01$ to $0.15 \pm 0.01$—and in the similarity index, which increased from $0.70 \pm 0.01$ to $0.945 \pm 0.007$.

To tackle more complex, continuous occlusions, we extended this approach to print a ball-in-cage model (Fig. 3e). In this more challenging scenario, the cage creates multidirectional occlusions around the central sphere. Unlike our previous approach, in which we parametrically generated the representative occluding surfaces using the scanned point cloud data, here we instead combined our auto-alignment workflow to precisely match the a priori reference model to the scanned point cloud, thus aligning, rather than generating, the occluding geometry (Supplementary Fig. 11). The aligned mesh was used as the occlusion input for the OSMO-based optimization.

Shadow-corrected models demonstrated statistically significant improvements, compared with uncorrected prints (Fig. 3f and Supplementary Methods 12), with a reduction in surface root mean square (r.m.s.) error from $0.50 \pm 0.09$ μm to $0.18 \pm 0.05$ μm and an increase in sphericity from $0.830 \pm 0.060$ μm to $0.965 \pm 0.006$ μm (Fig. 3g,h). Finally, we demonstrated enhanced printing of vessel-containing structures surrounded by a shadowing cage (Supplementary Methods 13 and Supplementary Fig. 12). These experiments highlight the suitability of GRACE not only for creating context-aware geometries but also for mapping occlusive features within the printing volume and mitigating their influence.

## Bioprinting with GRACE

Next we explored the potential of GRACE for biofabrication, a field aiming to produce engineered tissues for regenerative medicine, and as in vitro models for personalized medicine. Specifically, we aimed to fabricate adaptive geometries around living cells and organoids, showcasing: (1) the production and functionality of customized vessel-like channels; (2) the generation of automatically aligned multitissue prints; and (3) the compatibility of GRACE with further fabrication techniques. As a first demonstrator—inspired by how in vivo blood vessels grow to reach cells and provide nutrients—we assessed GRACE to automatically generate adaptive vessel-like networks optimized around dense cellular structures. Notable efforts in bioprinting focused on producing convoluted channel networks to nurture tissues[23–25]. However, none of these approaches can adaptively print channels reaching every cellular structure or organoid within the construct.

Here we combined GRACE with Embedded Extrusion-Volumetric Printing (EmVP)[3,26] to generate adaptive vascular-like architectures around toruses (extruded into GelMA) densely laden with insulin-secreting pancreatic cells (iβ-cells, $5.0 \times 10^7$ ml$^{-1}$; Fig. 4a and Supplementary Methods 14).

Networks around the toruses were generated with channel diameters of 1 mm at the inlets, tapering to 0.4 mm at the scaffold midplane, maintaining a fixed surface area of approximately $180 \pm 10$ mm$^2$ (Supplementary Fig. 13) and a 300-μm offset from the torus (Fig. 4b). To evaluate the efficacy of this approach, we compared these adaptive structures to two controls: randomly generated channels of the same surface area (Fig. 4c) and a bulk structure without channels as negative control. Following 24 h of dynamic culture, we observed a substantial increase in proinsulin secreted from GRACE-printed structures compared with both the random non-targeted and bulk controls ($3.2 \pm 0.3$, $2.0 \pm 0.4$

and $1.4 \pm 0.1 \times 10^5$ relative light units, respectively) (Fig. 4d). This result suggests superior mass transport within the adaptive geometries, probably because of the improved proximity of the cells to the surface of the channels. This therefore results in shorter diffusion distances when compared with randomly distributing channels throughout the whole construct.

Next, through automated alignment, a two-component cell-laden bone and cartilage model was produced. We prepared two GelMA-based resins containing articular cartilage-derived progenitor cells (ACPCs, $1.0 \times 10^7$ ml$^{-1}$) and bone-marrow-derived mesenchymal stem cells (MSCs, $5.0 \times 10^6$ ml$^{-1}$; Supplementary Methods 15). Femur models were first printed within the MSC-laden GelMA, washed, then placed within the printing vat filled with the ACPC bioresin. The MSC component was scanned to determine the orientation and position of the femur. The cartilage phase was then automatically positioned and printed to form a layer around the femoral head, building a native-like osteochondral architecture (Fig. 4e). Cells remained functional and persisted in their intended compartment (chondral or osteal) over 4 weeks, with ACPCs and MSCs synthesizing cartilage and mineralized bone matrix components, respectively, as confirmed by means of histology (Fig. 4f, Supplementary Fig. 14 and Supplementary Methods 16).

Finally, we demonstrated the versatility of GRACE by using an alternative printing modality: filamented light (FLight) biofabrication, a vat polymerization technique that generates structures composed of multicentimetre-long aligned microfilaments[16]. Here two differently stained populations of MSC spheroids (Supplementary Methods 17) were identified within the vat and the parametric model encased each group in a unique shape (stars or circles for green-stained or red-stained spheroids; Fig. 4g–j), resulting in the formation of elongated filamentous constructs spanning the width of the print volume (Fig. 4h–j). This showcases the ability of GRACE to adapt to different printing modalities while maintaining its core capability of generating context-aware, population-specific geometries.

## Discussion and outlook

This study introduced GRACE, an innovative workflow that makes use of the unique attributes of volumetric printing to generate context-aware geometries. By integrating light-sheet microscopy, computer vision algorithms and parametric modelling, GRACE can detect and respond to features across several scales, from organoids to macroscopic structures, and rapidly fabricates complex geometries that dynamically adapt to arbitrarily distributed features within the print volume. This level of automated, context-driven fabrication would be prohibitively time-consuming and impractical with manual design. GRACE operates with minimal user intervention, requiring only experiment-specific adjustment of the parametric models (Supplementary Fig. 6), thus greatly streamlining the fabrication workflow while simultaneously expanding the complexity and functionality of achievable structures. The versatility extends beyond its current implementation, as the concept could be integrated into other printing approaches. Examples include xolography, which inherently uses light-sheet optics[27], multiphoton printing, acoustic-based printing[28,29] or extrusion printing in suspension baths[30]. As the parametric models are agnostic to the imaging method generating the input data, alternative scanning modalities could be explored. For instance, optical tomography, or holographic approaches[31,32], for non-fluorescent imaging, could provide a broader safe photoexposure window, although we did not see any relevant phototoxic effects within our experiments (Supplementary Methods 18 and 19 and Supplementary Figs. 15 and 16). In terms of future directions, the ease of performing overprinting offered by GRACE could aid several innovative applications, for example, in soft robotics, for introducing more refined polymeric skins onto previously printed movable parts or accurately controlling the geometry of hydrogel-based osmotic actuators overprinted across skeletal-like scaffolds[22].

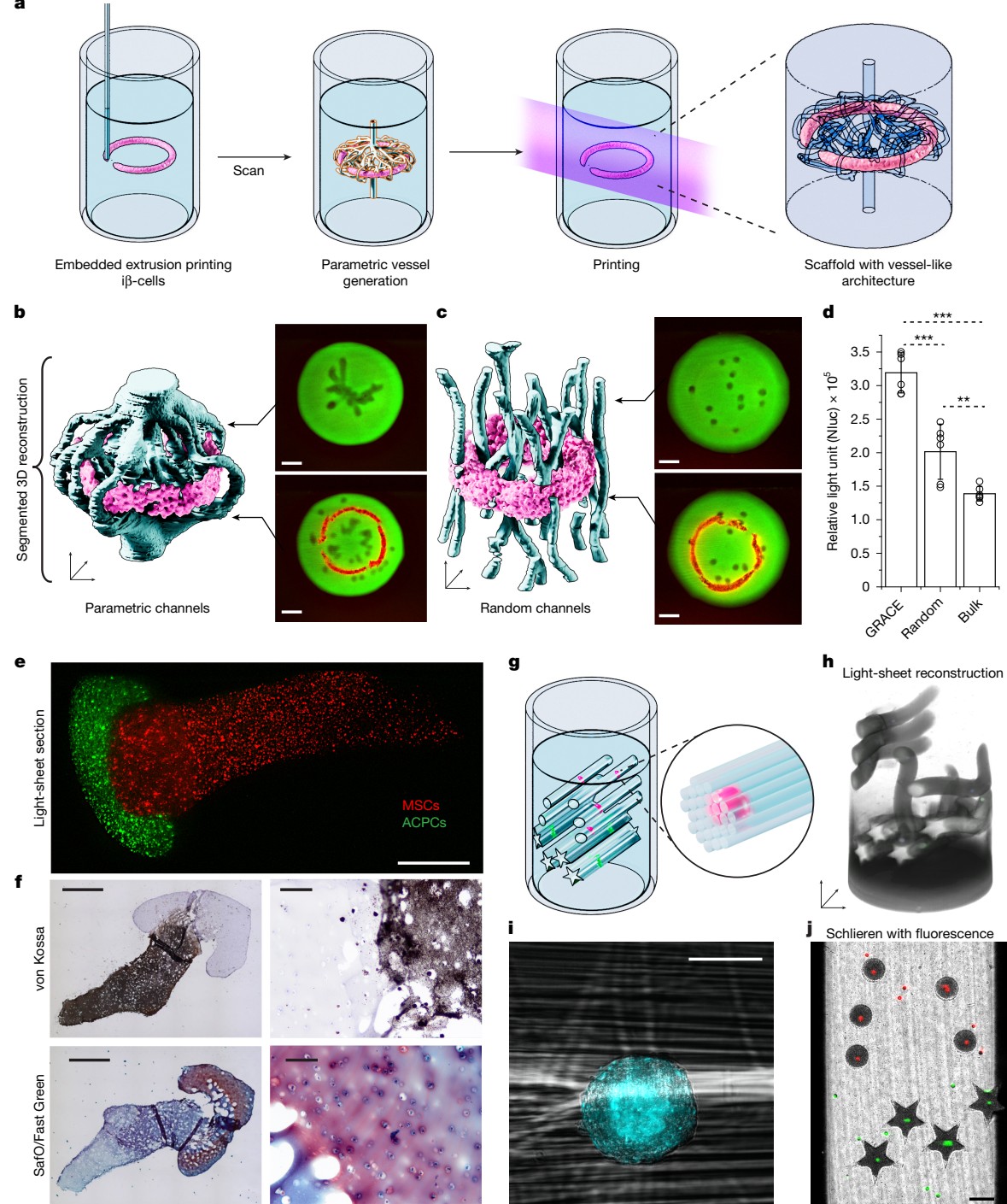

**Fig. 4 | Bioprinting of functional living tissues with cell-location-driven features enabled by GRACE. a**, Diagram of experiment process depicting the scanning of the volume, feature-driven model generation around the torus and printing of the scaffold. **b,c**, 3D segmentations of the resultant structures printed around the extruded toruses, both targeted and random, also showing light-sheet micrographs of the scaffold cross-sections at two planes. Scale bars, 2 mm. **d**, Graph showing the amount of insulin stored and released by iβ-cells by means of the bioluminescent reporter NanoLuc (Nluc) for the GRACE print, as well as for the random and bulk controls (mean ± s.d., ANOVA, *n* = 6, **\**P* < 0.01, **\***P* < 0.001, *F* = 55.74). **e**, A light-sheet section of the construct immediately following printing with the automatically aligned femur-cartilage model showing distinct osteal and chondral regions. Scale bar, 2 mm. **f**, The

histological sections following 4 weeks of maturation highlight the presence of a mineralized compartment (von Kossa positive, brown/black staining) and of glycosaminoglycans-rich cartilaginous tissue (SafO, red staining) (*n* = 3). Scale bars, 500 μm (left panels); 200 μm (right panels). **g**, 3D model of FLight construct generated with GRACE after scanning. Printing was performed around a subset of each spheroid population to avoid overcrowding. **h**, Resultant light-sheet 3D reconstruction after printing and washing. **i**, Filamentous construct is visible surrounding the spheroid. Scale bar, 100 μm. **j**, Discrimination of spheroids is evident in the schlieren image (captured immediately after printing) with fluorescence overlay, with the spheroids precisely positioned through the central axis of each star or circle FLight structure. Scale bar, 2 mm.

The GRACE workflow has direct implications for biofabrication, with the creation of biomimetic scaffolds that can adapt to the spatial distribution of cells or organoids, and for the fabrication of tissue constructs with highly controlled architecture, relevant to regulate cell function and tissue maturation. These systems could already serve as models for biomedical and pharmaceutical research. Although light-sheet imaging at present enables scanning of multicentimetre-sized volumes[33], future work would be necessary to produce constructs having (human) full-tissue-scale sizes. For instance, combining light-sheet imaging and GRACE with movable vats for volumetric printing could expand the imaging and fabrication range, allowing a form of mosaicking the sample volume[34]. Moreover, with larger imaging and printing ranges, new techniques to mitigate scattering will become increasingly needed[35–37]. In parallel, much progress has been made in designing self-assembly materials that can be used in bioprinting, which allow to tune the cellular microenvironment at the (sub)cell-level scale[38]. Converging these classes of materials with GRACE could further permit to better approximate the hierarchical composition of living tissues at the macro-to-micro scale (through printed architecture), down to the micro-to-nano scale (provided by the structured materials). GRACE also opens new avenues for adaptively modifying an object at any time point post-printing. Modifications could include spatial-selective grafting of biomolecules[39] and modulating stiffness gradients[40] or viscoelasticity[41]. Altogether, these advancements underscore the versatility of GRACE and its potential in both general additive manufacturing and bioprinting contexts, representing a shift in how printing is performed.

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

# Methods

## Tomographic volumetric printer

The custom-built volumetric printer used in this study (Fig. 1a) used a 405-nm laser source shaped into a flat-top intensity profile through coupling of the beam into a square core fibre (WF 70 × 70 μm, Ceram-Optec). This shaped beam was collimated and used to illuminate a DMD (Hi-Speed V-7000, ViALUX), for which the resultant projection was Fourier filtered and imaged with 1:1 magnification using a $4f$ relay onto the centre of the printing volume in a telecentric projection regime. During the printing process, vials were immersed within a refractive index compensating bath comprising a square water-filled quartz cuvette (OP36, eCuvettes). This was required to minimize refractive errors arising from the curvature of the vial surface, for both printing and feature scanning. To accommodate different print requirements, we used borosilicate glass vials (Readily3D) with internal diameters ranging from 8.8 to 15.3 mm, selected according to the specific model parameters and target print volumes. In terms of compatible resins, tomographic volumetric printing is compatible with processing light-crosslinkable materials, whereas other classes of materials can be added as inclusions in the resin vat, for instance, by means of embedded extrusion printing[3], before polymerizing the resin.

## Light-sheet imaging module

The light-sheet imaging path (Fig. 1) used three 40–50-mW diode lasers operating at 450 nm (RLDD450-40-5, Roithner), 532 nm (RLDD532-50-3, Roithner) and 650 nm (RLDH650M-40-5, Roithner) to cover a broad range of commonly used fluorophores. The beams were combined with dichroic mirrors, shaped into a flat-top fan profile with a 30° Powell lens (43-473, Edmund Optics) and then focused to the vial centre using an $f$ = 50 mm cylindrical lens. Two scanning regimes were used: a half-sweep mode covering $\Theta_{total} = \pi$ (500 polar sections) and a $\Theta_{total} = 2\pi$ sweep mode (1,000 polar sections). The latter was preferentially used for samples exhibiting occlusive or highly scattering samples, as this provided bidirectional illumination of the features. Image acquisition for feature scanning/registration was performed using a monochromatic camera (Alvium 1800 U-240m, Allied Vision) in conjunction with an $f$ = 50 mm C-Mount lens (MVL50M23, Thorlabs). With this hardware, we achieved a resolution of 14.47 μm along the image plane, suitable for capturing cellular aggregates and organoids. Higher-resolution hardware would be needed to extend the detection to the single-cell regime, hence the current system is limited to detecting larger particles. Exposure and gain parameters were manually set before each scanning session. A rotary filter mount containing band-pass filters for several commonly used fluorophores (GFP, Cy3.5, Cy5) was integrated post-objective to facilitate spectral selection of the emission signal while rejecting the backscattered laser line. Notably, although light-sheet imaging has been historically an expensive technology, new reports describe how to build open-source, affordable light-sheet systems. In our case, our simplified set-up required only the use of a laser source, a beam reducer, a Powell lens for beam shaping and a cylindrical lens for focusing the light sheet within the volume—all available as off-the-shelf components.

## Image processing and feature registration for GRACE

A MATLAB script was prepared to perform several functionalities key to the GRACE workflow. This process (shown in Extended Data Fig. 1a, and detailed extensively in Supplementary Methods 4) was responsible for the following: (1) the initialization and synchronization of hardware (imaging, light sheet and printing) and software parameters; (2) the acquisition and processing of polar light-sheet image stacks; (3) the isolation and registration of features of interest from the background; (4) the conversion of image data to usable 3D coordinate data; (5) the processing of these data (for example, using clustering detection) to extract construct-specific information necessary for generating

the desired construct; and (6) outputting these data for use with the parametric modelling software.

## Data-driven parametric models

We used off-the-shelf software Rhino3D (Robert McNeel & Associates) in conjunction with its integrated Grasshopper (GH) visual programming environment. GH definitions were developed to create parametric models for each geometry type. These definitions performed three key tasks: (1) importing and synchronizing with data files containing coordinates, radii and other relevant data as described above; (2) using these data to generate the desired parametric geometry; and (3) baking and exporting the final model as an STL file suitable for 3D printing. Our work focused on three broad types of biologically inspired geometry: perfusable vessel-like channels surrounding scanned features with an inlet and an outlet, positive interconnected geometries and targeted single-layer encapsulation. In several of these cases, although not strictly necessary, we also used two freely available add-ons for GH. These were the Dendro plug-in[42], which facilitated convenient surface generation around point structures and the ShortestWalk plug-in[43], which makes use of the A* algorithm[41] to calculate the shortest walk in a network of paths. We note that similar functionalities can be provided using other plug-ins (either inbuilt or third party) or by writing a custom script within GH. See Supplementary Methods 8 for further details on the creation of the parametric models and definitions.

## Synthesis of GelMA

All chemicals were obtained from MilliporeSigma and used without further purification or modification, unless stated otherwise. GelMA was synthesized as previously reported[44]. Briefly, 0.6 g of methacrylic anhydride were added per gram of gelatin (type A, from porcine skin, 10 w/v% in phosphate-buffered saline (PBS)) and left to react for 1 h at 50 °C under constant stirring, to obtain a degree of methacryloyl substitution of 80%, as assessed through $^1$H-proton nuclear magnetic resonance (NMR; 400 MHz, Agilent 400-MR NMR, Agilent Technologies). The resulting solution was dialysed (MW cut-off = 12 kDa) against deionized water to remove the unreacted methacrylic anhydride. The purified macromer solution was sterile filtered (0.22 μm), freeze-dried and stored at −20 °C until used.

## Resin preparation and printing

GelMA-based bioresin supplemented with 0.1% w/v LAP (Tokyo Chemical Industry) photoinitiator was used for experiments in this work. Before printing, resins were thermally gelled by immersing in ice water for 10 min. Schlieren imaging was used to observe the crosslinking process during printing, for which the process was manually stopped on observing a sufficient accumulation of dosage, as evidenced by a rapid change in the refractive index of the resin. After printing, samples were washed with warm (37 °C) PBS to remove uncrosslinked material. The mechanical characterization of the printed hydrogels is reported in Supplementary Methods 20 and Supplementary Fig. 17.

## Linear light-sheet scanning of samples post-printing

For all experiments in this paper, printed samples were imaged using a custom-built, linearly swept light-sheet fluorescence microscope. Samples were placed in a 22 × 22-mm square cuvette (OP36, eCuvettes) and immersed in PBS during imaging. Sections were captured at 35-μm steps over the span of the entire sample.

## Using GRACE to detect and print around alginate particles

For all work involving alginate microparticles (microparticle preparation is described in Supplementary Methods 6), resins were prepared with 10% GelMA w/v + 0.1% LAP w/v. The stained alginate particles were laden into the volume and gently suspended by agitating the volume while being cooled in an ice bath until the resin thermally gelled. These were then scanned over $\Theta_{total} = \pi$ and cluster detection

was used to determine the centroid coordinates of each alginate sphere (as described above in the pipeline for feature coordinate registration in GRACE). An example of detecting and keeping track (through selective illumination) of a particle is depicted in Supplementary Video 3. For prints involving two populations of stained alginate particles, this process was performed twice to obtain a separate coordinate dataset for each group. The corresponding adaptive models were then generated and printed according to the GRACE workflow. This approach could also be performed in the presence of alginate capsules, each loaded with a concentration of 200 million cells per ml (Supplementary Fig. 18, Supplementary Information Methods 21 and Supplementary Video 4).

## Auto-alignment of sequential prints

Auto-alignment of sequential prints was accomplished using a combination of GH and MATLAB. In our work, this was demonstrated with a femur and cartilage model, although the same workflow can be performed to automatically align any two or more geometries sequentially. The process began by importing a reference femur model into GH, in which a section of geometry from the femoral head was extracted and assigned an adjustable extruded thickness, allowing for customizable cartilage thickness. The relative positioning between the two models defined the target alignment between the sequential geometries a priori. To accomplish this alignment practically, we sought to determine the rigid transformation required to align the reference femur (and thus the relatively placed cartilage model) to the location of the arbitrarily located scanned femur within the vial. First, a stained femur (Cy5) was printed using 10% GelMA w/v + 0.1% LAP w/v, washed and resuspended into new GelMA resin (stained with Cy3.5) at a random orientation. This print was then scanned, and the resulting volumetric point cloud data $P_{scan} = \{q_1, q_2, ..., q_m\}$ was exported. A dense array of random points $P_{Ref} = \{p_1, p_2, ..., p_n\}$ was then generated within the volume of the reference femur model and then also exported to MATLAB. The reference femur point cloud ($P_{ref}$) was automatically aligned to the scanned femur ($P_{scan}$) using the iterative closest point (ICP) algorithm (MATLAB's 'pcregistericp' function). This generated a rigid transformation matrix $T$, such that, when also applied to the reference cartilage geometry $G_{Ref}$, resulted in a rotation and translation of the cartilage component to its correctly aligned position over the scanned femoral head within the vial, such that $G'_{Aligned} = TG_{Ref}$. The transformation data were synchronized to the GH definition, thus generating a correctly placed cartilage geometry that was subsequently exported and printed. By using any reference models and assigning correct relative positions within GH, this process can be used for any arbitrary alignment task, providing a versatile approach for complex, multistage printing processes that require precise spatial relationships between sequentially printed components. Notably, once the reference geometries and relative positions were pre-established, the entire alignment process could be accomplished <15 s after scanning. Additional information on auto-alignment and overprinting applications is included in Supplementary Methods 22 and 23.

## Shadow correction of pillar occlusions

An occluding structure comprising ten parallel pillars 0.5 mm in diameter attached to a base was fabricated using a Formlabs Form 3B+ stereolithography (SLA) printer with opaque grey resin (Formlabs Gray Resin V4) (Fig. 3c, Supplementary Fig. 10a and Supplementary Methods 11). The occluder was placed into vials and filled with 1 ml of 10% GelMA w/v + 0.1% LAP w/v. The GRACE workflow was performed as previously described but without emission filters during scanning. Instead, the reflected and scattered laser line was imaged over a $\Theta_{total} = 2\pi$ sweep, enabling approximate surface reconstruction (Supplementary Fig. 10b) by using the light sheet as a profilometer. Cluster detection was used on the surface data to identify the centroids and a principal component analysis determined the pitch and yaw of

the pillars (Supplementary Fig. 10c and Supplementary Methods 11 for details). A parametric model was prepared in GH to generate the representative occludsion to overlay with the scanned data at the correct angles and locations. This 3D model of the occlusion, along with a symmetrical cog model, was exported for processing using the OSMO algorithm[21] to create an optimized set of shadow-corrected projections for printing. It should be noted that the shadow correction algorithm is not designed to prevent or mitigate artefacts caused by (unwillingly) introducing bubbles within the resin vat. As such, careful handling and gentle pipetting or pouring of the resin is always recommended when loading the printable materials into the vials. Moreover, the efficacy of the corrective algorithm reduces proportionally to the number and size of occluding elements present in each plane within the print volume, as evidenced by the analysis reported in Supplementary Fig. 9. Because the exact number, shape and size of occluding elements that can be tolerated for printing are not constant and depend on the architecture to be printed, it is advisable to perform in silico simulations, as reported in Methods, ahead of printing experiments, to evaluate printability.

## Shadow correction for ball-in-cage model

A spherical cage-like occluding structure was fabricated using stereolithography (Fig. 3e), following the same protocol and material as with the occluding pillars. The geometry was algorithmically generated in GH through a three-step process: (1) 45 nodes were randomly distributed on the surface of a 10-mm-diameter sphere; (2) for each node, paths were computed to its five nearest neighbours based on spatial proximity; (3) struts of 0.5 mm diameter were generated along these paths, connecting each node to its five nearest neighbours. This procedure yielded a complex, non-uniform, interconnected spherical cage structure (Supplementary Fig. 11a), presenting a challenging occlusion scenario for volumetric printing. The cage was embedded within the resin, thermally gelled and scanned using the light sheet as a profilometer, as previously described. This scan generated a sparse point cloud representing the occluding surface (Supplementary Fig. 11b). In contrast to the pillar experiment, in which the complete occlusion geometry was reconstructed parametrically, here we instead used the auto-alignment protocol previously developed for the femur-cartilage model. A reference cage mesh was algorithmically aligned to the scan-derived point cloud, enabling the correctly oriented reference mesh to serve as the computational occlusion volume during projection optimization using the OSMO algorithm. A 5-mm-diameter sphere positioned within the cage served as the target geometry for optimization and volumetric printing. Post-printing, the corrected and uncorrected structures (printed in triplicate) were washed, removed from the occluding cages and imaged using linear light-sheet microscopy (Supplementary Fig. 11c). Prints were quantified on the basis of their r.m.s. error and sphericity (Fig. 3f–h; see Supplementary Methods 12 for further information). Moreover, using this same shadow correction and printing approach, a more complex trifurcated vascular network was printed within a hydrogel enclosed into a stent-like occluding mesh, to demonstrate the possibility to resolve also hollow and negative features (Supplementary Methods 13, Supplementary Fig. 12 and Supplementary Video 5).

## iβ-cells subculture and expansion

iβ-cells, an engineered pancreatic cell line mimicking β-cell function and capable of releasing insulin together with the luminescent reporter NanoLuc, were obtained as previously described in the literature[26,27]. iβ-cells were cultured (95% humidified incubator at 37 °C, 5% $CO_2$) in Roswell Park Memorial Institute (RPMI) 1640 Medium, containing GlutaMAX and HEPES (Gibco, Life Technologies) supplemented with foetal bovine serum (FBS, 10% v/v), 100 U ml$^{-1}$ penicillin and 100 µg ml$^{-1}$ streptomycin. For all experiments, cells were used at passages 3 and 4.

## GRACE printing within constructs produced by embedded extrusion

A toroidal extrusion path of 4 mm in diameter was designed and saved as a G-code file. First, the volumetric printing vial was loaded with 5% GelMA at 37 °C + 0.1% w/v LAP. The vial was left to thermally gelate at room temperature overnight and then placed in a R-GEN 100 extrusion printer (RegenHU) and centrally retained using a custom bracket. The GelMA hydrogel was used as a photoreactive suspension bath for embedded extrusion printing. Extrusion printing was performed with the R-GEN 100 printer by means of a pneumatic-driven extrusion printhead. As bioink for extruding the toruses, iβ-cells ($5.0 \times 10^7$ cells ml$^{-1}$, stained with the Vybrant DiD membrane dye (ThermoFisher Scientific) to facilitate imaging) were suspended in a 2% w/v alginate solution, used as fugitive viscosity enhancer to improve printing resolution of the high-density cell suspension. The bioink was loaded into 3-ml cartridges equipped with a 23 G stainless steel straight, cylindrical needle (Nordson EFD). Toruses were then extrusion-printed in sterile conditions within the GelMA support bath. After this, the vial containing the support bath and the cell-laden bioink was transferred to the volumetric printer. Following the GRACE workflow, the samples were imaged by means of light sheet and a set of blood-vessel mimetic channel networks were parametrically generated by the software to wrap around the cell-laden toruses (Fig. 4a and Supplementary Methods 14). Using tomographic volumetric printing, the design was printed into the GelMA bath, forming the biological tissue construct. This produced tapered channel networks around the toruses with a 300-μm offset, fixed surface area of $180 \pm 10$ mm$^2$, minimum diameter of 450 μm and an inlet and outlet of about 1 mm at both ends of a cylindrical bulk scaffold. After volumetrically printing these constructs, the vial was heated to 37 °C to dissolve the unpolymerized GelMA and the sample was retrieved and washed with prewarmed PBS. It should be noted that, although the CAD design of the extruded cell structures is known, it is still preferred to apply the GRACE algorithm after 3D imaging of the extruded features and not directly on the geometry from the CAD file. Generating the parametric design after imaging, in fact, ensures that any printing artefact, or loss of shape fidelity owing to the viscoelastic nature of the cell-based ink, is taken into account[45] (Supplementary Fig. 19). Moreover, this also takes advantage of the ability of GRACE to auto-align the volumetric print onto the features of interest, reducing inaccuracies and errors that could be caused by the manual alignment of the print vial. Furthermore, two control groups were produced: (1) a bulk cylindrical structure containing no channels printed around the extruded toruses as negative control and (2) a set of randomly generated channels running along the length of the cylindrical bulk, also maintaining the same $180 \pm 10$ mm$^2$ surface area, to provide a comparable interface for solute exchange from the printed vessels to the hydrogel. Equal surface areas of the random and GRACE-printed constructs were verified post-printing by using linear light sheet to scan the samples and then manually segmenting the channels and determining the surface areas. From these data, the average minimum torus to vessel distance and the average vessel diameter were calculated (Supplementary Fig. 13).

## Analysis of embedded extrusion prints

The printed constructs ($n = 6$ for each group) with different generated vessel networks were retrieved and cultured overnight in incubator in the presence of RPMI 1640 medium, GlutaMAX, HEPES (Gibco, Life Technologies) supplemented with 10% v/v FBS and 1% penicillin/streptomycin. Dynamic culture was permitted by placing the samples on an orbital shaking platform (95 rpm), to ensure media displacement and flow. The following day, the supernatant was collected for each different condition and the bioluminescent reporter NanoLuc (directly related to the amount of insulin stored and released by iβ-cells) was quantified using the NanoLuc Luciferase Kit (Promega Corporation) against a standard curve, using a CLARIOStar Plus multimodal plate reader (BMG Labtech) (Fig. 4d).

## Osteochondral differentiation in the cell-laden femur-cartilage model

At passage 4 (see Supplementary Methods 15 for cell isolation and expansion protocol), ACPCs and MSCs were encapsulated in 10% w/v GelMA solution in PBS at a final density of $1.0 \times 10^7$ cells ml$^{-1}$ and $5.0 \times 10^6$ cells ml$^{-1}$, respectively. LAP dissolved in PBS at 0.1% w/v was used as a photoinitiator for the crosslinking. To minimize scattering owing to the presence of cells, 30% v/v of iodixanol (OptiPrep) was used in the MSC-containing resin, whereas a 20% concentration was used for resin containing ACPCs. Vybrant DiD and DiO (ThermoFisher Scientific) were used as cell-labelling membrane staining for MSCs and ACPCs, respectively. Cells were stained for 30 min at 37 °C according to the manufacturer's instructions. Bone-cartilage models were scanned, aligned and printed with GRACE (Fig. 4e), in the same fashion as described previously with acellular constructs. Post-printing, the cell-laden constructs containing both the bone and cartilage layers were washed in warm PBS to remove the unpolymerized biomaterial and cultured in 1:1 ratio osteogenic (Osteogenic Differentiation Medium Bullet-Kit, PT-3002, Lonza) and chondrogenic media (DMEM, supplemented with 100 U ml$^{-1}$ penicillin, 100 μg ml$^{-1}$ streptomycin, 0.2 mM L-ascorbic acid-2-phosphate, 1% v/v Insulin-Transferrin-Selenium (ITS) + Premix Universal Culture Supplement (Corning), $0.1 \times 10^{-6}$ M dexamethasone and 10 ng ml$^{-1}$ recombinant human TGF-β1) for 4 weeks. Media were refreshed three times a week.

## Combining FLight with GRACE

Each stained population of spheroids (see Supplementary Methods 17 for MSC spheroids preparation protocol) was scanned using either an excitation wavelength of 450 nm (for DiO) or 650 nm (for DiD), in conjunction with the appropriate emission band-pass filter. The centroid coordinates of each spheroid population were determined and synchronized with a parametric model. This model was designed so as to position either a cylindrical or a star-shaped geometry centred at each spheroid, with DiO-stained features receiving circles and DiD receiving stars. Following model export and system homing, we used FLight fabrication (Fig. 4g,h,j). This involved projecting a single binary image of the geometry at its face-on angle into the stationary vial, with the rotational stage fixed at the corresponding angle of the projection. Illumination was maintained until crosslinking was observed through approximately 70% of the vial volume, as determined by real-time schlieren imaging. The uncrosslinked material was then washed in warm PBS, linear light sheet imaged in the vial and then collected and imaged under confocal microscopy (Fig. 4i).

## Statistical analysis

Data are expressed as mean ± standard deviation (s.d.), with a minimum sample size of $n \geq 3$. Statistical analyses for experiments involving shadow correction were conducted using OriginPro 8.5 (OriginLab) and Prism 9 software (GraphPad Software Inc.) was used for the EmVP experiments. Normal distribution was assumed. Pairwise comparisons between two groups were performed using a Student's two-sample two-tailed $t$-test ($\alpha = 0.05$), with statistical significance defined as $P < 0.05$. For analyses involving more than two groups, a one-way analysis of variance (ANOVA) was used with post hoc Turkey's multiple comparison test.

## Ethical statement

Animal tissue and cells used in this study were obtained from an existing, commercially available cell line, as described in Methods, or deceased equine donors, donated to science by their owner, and according to the guidelines of the Institutional Animal Ethical Committee of Utrecht University.

## Reporting summary

Further information on research design is available in the Nature Portfolio Reporting Summary linked to this article.

## Data availability

The data that support the findings of this study are available in the manuscript, its Supplementary Information and source data files. The source data are also available at https://zenodo.org/records/16024214 (ref. 46). Data are also available from the corresponding authors on request. Source data are provided with this paper.

## Code availability

The scripts used for the workflow are available from the corresponding author on request.

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

**Acknowledgements** This work was performed at the Regenerative Medicine Center Utrecht, the Netherlands. We thank J. Burdick for the constructive feedback on the manuscript. This project received support from the European Research Council (ERC) under the European Union's Horizon 2020 research and innovation programme (grant agreement no. 949 806, VOLUME-BIO). R.L. and J.M. acknowledge the support from the Gravitation programme 'Materials-Driven Regeneration' financed by the Netherlands Organization for Scientific Research (024.003.013). R.L. acknowledges financial support from the Dutch Research Council's Talent programme (Vidi, 20387).

**Author contributions** Conceptualization: S.F., R.L. Methodology: S.F., G.G., D.R., M.G., R.L. Investigation: S.F., G.G., D.R., A.L. Visualization: S.F., G.G., D.R., A.L., M.G., E.G., R.L. Funding acquisition: R.L. Project administration: R.L. Supervision: R.L. Writing – original draft: S.F., G.G., R.L. Writing – review and editing: S.F., G.G., D.R., A.L., M.G., J.M., R.L. G.G. and D.R. contributed equally to this work.

**Competing interests** S.F. and R.L. are inventors on a provisional patent application PCT/NL2024/050642 that covers part of the workflow reported in this manuscript. R.L. is scientific advisor for Readily3D SA. The other authors declare no competing interests.

**Additional information**
**Correspondence and requests for materials** should be addressed to Riccardo Levato.

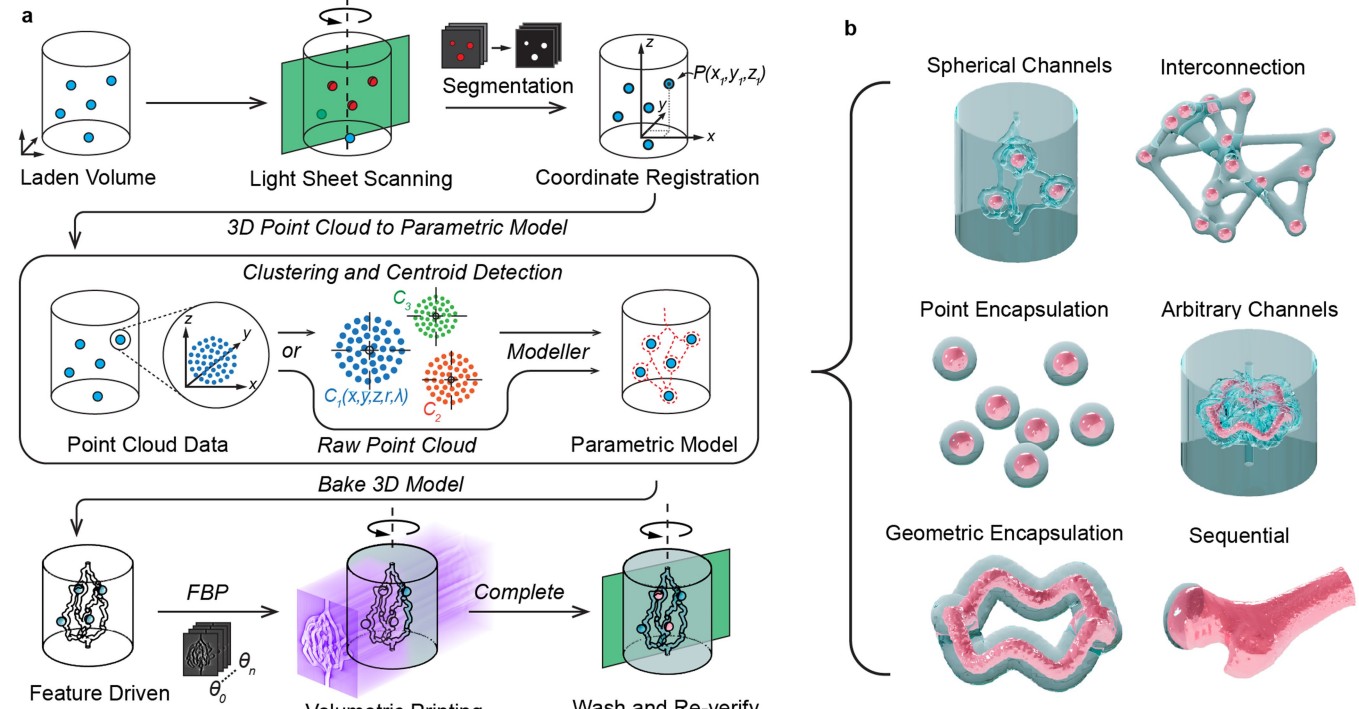

**Extended Data Fig. 1 | Flow chart illustrating the GRACE workflow. a**, Flow chart of the GRACE methodology. Top row: a printable volume, laden with features is presented; the volume is then scanned with light-sheet imaging and, through segmentation and image processing of the polar stack, the volumetric coordinates of the features is obtained in Cartesian coordinates. Middle row: the resultant point cloud is then either further processed using clustering detection to index and determine the centroids of each feature or alternatively used as-is—in all cases being sent to the parametric modeller. The parametric modeller uses the data to generate a bespoke model driven by and adapted to the scanned features. Bottom row: the parametric model is baked and the resultant file is then processed using the typical tomographic volumetric printing routines to generate a series of back-projections, which are then sent to the printer. The model is printed and washed, thus revealing the GRACE-printed structure. If desired, the printed structure can then be light-sheet imaged again for re-verification purposes. **b**, A subset of various positive and negative parametric geometries (shown in blue), generated using point-like and bulk scanned features (shown in red). Depicted are spherically wrapped channel networks around feature centroids, spherical features interconnected with struts, encapsulation of point-like structures, channel network creation around arbitrarily shaped features, geometric encapsulation of arbitrarily shaped features and sequential generation of parametric geometry in relation to another. This highlights the versatility of parametric modelling in generating a wide range of complex, feature-driven geometries.

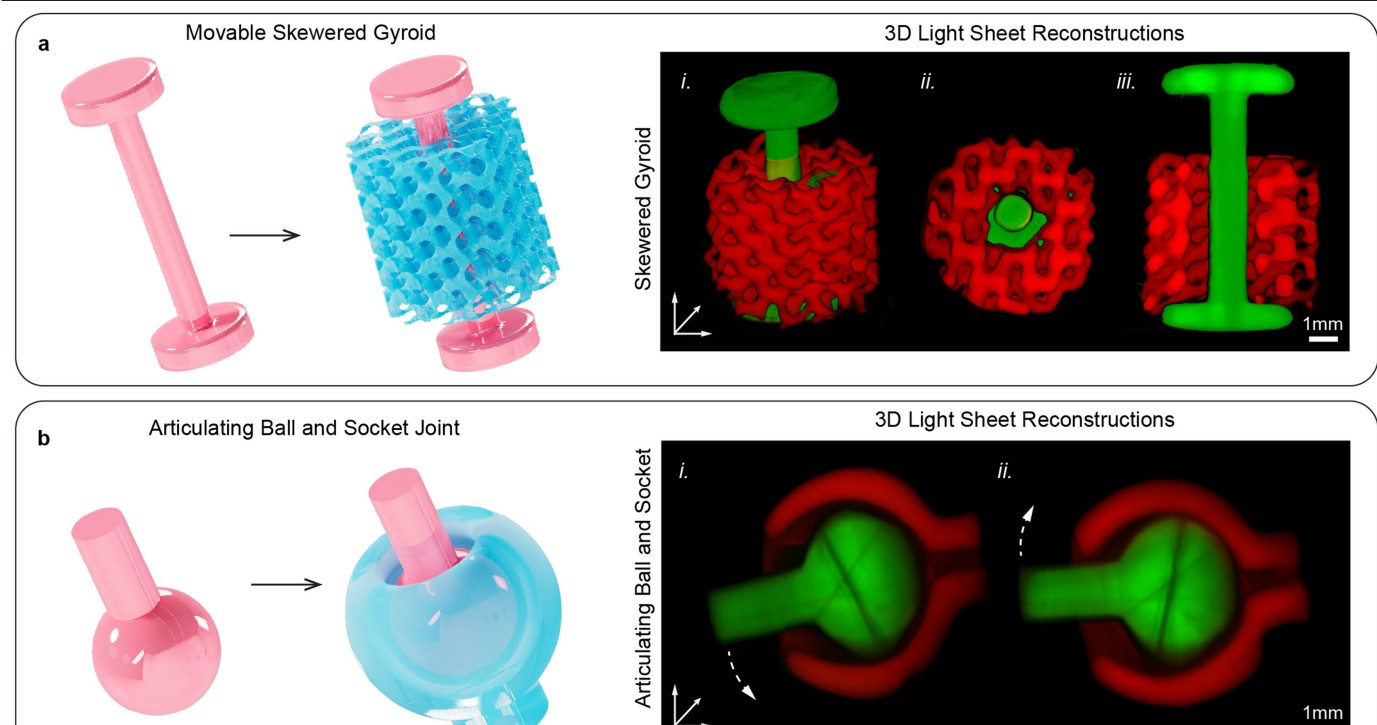

**Extended Data Fig. 2 | Articulating auto-aligned sequential prints with GRACE. a**, A gyroid with internal aperture (blue) is precisely aligned to the axis of a randomly oriented stud-like print (pink), thus forming a sliding cylindrical joint along the shaft. Light-sheet reconstructions post-printing and washing are shown, highlighting a perspective view (i) and half-section views through the top and side planes (ii and iii), clearly showing the central aperture of the gyroid. Note that, as imaging was performed with the construct standing vertically, the gyroid has visibly slid down the shaft, further indicating two distinct components. **b**, An automatically aligned articulating ball-and-socket joint, with the ball component (pink) printed first and then randomly resuspended. A socket was then automatically aligned to it through GRACE. The light-sheet reconstructions (i and ii) of the printed components show the articulating ball arm in two different positions, indicated by the arrows.

**Extended Data Table 1 | Table of key process durations with the various modalities of GRACE explored within our work**

| GRACE Printing | Duration [s] | Additional Info. |
|---|---|---|
| Light sheet scan ($\Theta_{total} = \pi$) | 75-150 | Double for $\Theta_{total} = 2\pi$. Dependent on signal exposure time per slice. |
| Feature isolation & coordinate export | 10-30 | Dependent on feature quantity. |
| Parametric model generation & export | 1-10 | Dependent on parametric model. |
| Back-projection generation | 60-80 | Ram-Lak filtered tomographic back projection. |
| Printing | 20-40 | Model and input power dependent. Multiply above by number of different spectral channels used. |

| GRACE Printing with Sequential Auto-alignment | Duration [s] | Additional Info. |
|---|---|---|
| Light sheet scan ($\Theta_{total} = \pi$) | 75-150 | Double for $\Theta_{total} = 2\pi$. Dependent on signal exposure time per slice. |
| Feature isolation & coordinate export of pre-printed geometry. | 10-30 | Dependent on feature quantity. |
| Generation of point cloud within reference model to facilitate alignment with scanned point cloud. | 10-20 | Only performed once and can be pre-calculated. |
| Alignment using iterative closest point algorithm. | 1-5 | Performed in MATLAB using 'pcregistericp' function. |
| Parametric model generation | 1-2 | Transformation of geometry to be printed, onto scanned geometry. |
| Back-projection Generation | 60-80 | Ram-Lak filtered tomographic back projection. |
| Printing | 20-40 | Model and input power dependent. |

| GRACE Printing with Shadow Correction | Duration [s] | Additional Info. |
|---|---|---|
| Light sheet scan ($\Theta_{total} = 2\pi$) | 160-240 | Double for $\Theta_{total} = 2\pi$. Dependent on signal exposure time per slice. |
| Occlusion mapping and export of raw surface point cloud. | 10-30 | Depending on occlusion size or quantity. |
| Parametric generation of occlusion surface (or alignment as above). | 1-10 | Dependent on parametric model. |
| Parametric occlusion model generation & export. | 5-10 | Transformation of geometry to be printed, onto scanned geometry. |
| Optional back-projection optimization of target model with occlusion using OSMO. Step used only in the experiments reported in Figure 3. | <7200 | OSMO Optimization with 20 iterations. |
| Printing | 20-40 | Model and input power dependent. |

| | |
|---|---|

# Reporting Summary

## Statistics

For all statistical analyses, confirm that the following items are present in the figure legend, table legend, main text, or Methods section.

| n/a | Confirmed | |
|---|---|---|
| ☐ | ☒ | The exact sample size (*n*) for each experimental group/condition, given as a discrete number and unit of measurement |
| ☐ | ☒ | A statement on whether measurements were taken from distinct samples or whether the same sample was measured repeatedly |
| ☐ | ☒ | The statistical test(s) used AND whether they are one- or two-sided<br>*Only common tests should be described solely by name; describe more complex techniques in the Methods section.* |
| ☒ | ☐ | A description of all covariates tested |
| ☐ | ☒ | A description of any assumptions or corrections, such as tests of normality and adjustment for multiple comparisons |
| ☐ | ☒ | A full description of the statistical parameters including central tendency (e.g. means) or other basic estimates (e.g. regression coefficient) AND variation (e.g. standard deviation) or associated estimates of uncertainty (e.g. confidence intervals) |
| ☐ | ☒ | For null hypothesis testing, the test statistic (e.g. *F*, *t*, *r*) with confidence intervals, effect sizes, degrees of freedom and *P* value noted<br>*Give P values as exact values whenever suitable.* |
| ☒ | ☐ | For Bayesian analysis, information on the choice of priors and Markov chain Monte Carlo settings |
| ☒ | ☐ | For hierarchical and complex designs, identification of the appropriate level for tests and full reporting of outcomes |
| ☒ | ☐ | Estimates of effect sizes (e.g. Cohen's *d*, Pearson's *r*), indicating how they were calculated |

*Our web collection on statistics for biologists contains articles on many of the points above.*

## Software and code

Policy information about availability of computer code

| Data collection | The following softwares were used for data collection:<br>- Matlab 2023a (MathWorks, USA)<br>- Rhino 7 (Robert McNeel & Associates, USA) |
|---|---|
| Data analysis | Statistical analyses for experiments involving shadow correction were conducted using OriginPro 8.5 (OriginLab, USA), while Prism 9 software (GraphPad Inc., USA) was used for the EmVP experiments. |

For manuscripts utilizing custom algorithms or software that are central to the research but not yet described in published literature, software must be made available to editors and reviewers. We strongly encourage code deposition in a community repository (e.g. GitHub). See the Nature Portfolio guidelines for submitting code & software for further information.

## Data

Policy information about availability of data

All manuscripts must include a data availability statement. This statement should provide the following information, where applicable:
- Accession codes, unique identifiers, or web links for publicly available datasets
- A description of any restrictions on data availability
- For clinical datasets or third party data, please ensure that the statement adheres to our policy

Supplementary Information is available for this paper. Correspondence and requests for materials should be addressed to r.levato@uu.nl. Animal cells used for this study and codes for the parametric printing process are available under a research agreement with Utrecht University.

## Research involving human participants, their data, or biological material

Policy information about studies with human participants or human data. See also policy information about sex, gender (identity/presentation), and sexual orientation and race, ethnicity and racism.

| | |
|---|---|
| Reporting on sex and gender | not applicable |
| Reporting on race, ethnicity, or other socially relevant groupings | not applicable |
| Population characteristics | not applicable |
| Recruitment | not applicable |
| Ethics oversight | not applicable |

Note that full information on the approval of the study protocol must also be provided in the manuscript.

# Field-specific reporting

Please select the one below that is the best fit for your research. If you are not sure, read the appropriate sections before making your selection.

☒ Life sciences   ☐ Behavioural & social sciences   ☐ Ecological, evolutionary & environmental sciences

For a reference copy of the document with all sections, see nature.com/documents/nr-reporting-summary-flat.pdf

# Life sciences study design

All studies must disclose on these points even when the disclosure is negative.

| | |
|---|---|
| Sample size | Sample size was at least n>=3 for all experiments. No power analysis was performed ahead of the study. The choice is based on the previous experiment in which the selected size were shown to be sufficient to observe statistically significant differences, for the magnitude of effect we typically observe. For all cell culture tests, sample size was either 3 or 4. For printing tests (cell free), samples were printed in 3 replicates. Monte Carlo simulations (data analyzed and reported in figure 2b) were repeated 12 times. |
| Data exclusions | no exclusion |
| Replication | Attempts at replications in our lab were successful |
| Randomization | not applicable |
| Blinding | not applicable. Investigators could not be blind when extracting data from imaging information (the images clearly reveal which experimental group is being analyzed). |

# Reporting for specific materials, systems and methods

We require information from authors about some types of materials, experimental systems and methods used in many studies. Here, indicate whether each material, system or method listed is relevant to your study. If you are not sure if a list item applies to your research, read the appropriate section before selecting a response.

## Materials & experimental systems

| n/a | Involved in the study |
|---|---|
| ☒ | ☐ Antibodies |
| ☐ | ☒ Eukaryotic cell lines |
| ☒ | ☐ Palaeontology and archaeology |
| ☒ | ☐ Animals and other organisms |
| ☒ | ☐ Clinical data |
| ☒ | ☐ Dual use research of concern |
| ☒ | ☐ Plants |

## Methods

| n/a | Involved in the study |
|---|---|
| ☒ | ☐ ChIP-seq |
| ☒ | ☐ Flow cytometry |
| ☒ | ☐ MRI-based neuroimaging |

## Eukaryotic cell lines

Policy information about cell lines and Sex and Gender in Research

| | |
|---|---|
| Cell line source(s) | We used iB cells, originally developed by the Fussenegger group at ETH Zurich, who also provided these cells to us. The papers describing how the line is produced are listed in the Methods section (specificlaly, reference 27: https://onlinelibrary.wiley.com/doi/10.1002/smll.202101939). |
| Authentication | cells are capable to release insulin (measured by ELISA) and nanoLuc (measured via luminescent assays, NanoLuc Luciferase Kit (Promega Corporation, USA)) as reporter. |
| Mycoplasma contamination | not tested |
| Commonly misidentified lines (See ICLAC register) | iB cells is a line derived from engineering 1.1E7 with synthetic gene circuits to release insulin and nanoLuc as reporter. While 1.1E7 a cell line listed in the ICLAC register, iB is a new line (see above), described by the Fussenegger lab at ETH Zurich. In this work, iB has been used to assess its capacity to release insulin and nanoLuc as a function of the bioprinted vascular geometries. The cells were able to perform this function, and therefore to be safely used for our experiment. |

## Plants

| | |
|---|---|
| Seed stocks | not applicable |
| Novel plant genotypes | not applicable |
| Authentication | not applicable |

