## [Peer Review File · Nature]

Adaptive and Context-Aware Volumetric Printing

Corresponding Author: Dr Riccardo Levato

Version 0:

Reviewer comments:

Referee #1

(Remarks to the Author)

Summary: The manuscript presents an innovative approach to Generative, Adaptive, Context-Aware 3D Printing (GRACE). The integration of volumetric imaging, parametric modeling, and adaptive fabrication is well-explored, demonstrating potential applications in bioprinting and materials science. However, several areas require further clarification, additional validation, and improved organization to enhance readability and scientific rigor.

Major Comments:

1. The goal of the researchers is well stated, to enable the printer to see the structure that is within the photoresin bath and to generate an additional print based on that first structure (printed or placed). However, in general while the data is a clear first step in that direction, the degree of progress appears overstated. The statement at the end of the introduction that "This approach is compatible with a wide range of fabrication technologies and materials, including complex composites, and living cells with bio-friendly hydrogels, unlocking a new generation of applications in tailor-made tissue mimetic materials and data-driven rapid prototyping." Is overstated, the presented data does not support this.
2. On an overall note, there are multiple occurrences of language used in the manuscript that lacks technical rigor and seeks to imply impact that is not clear or quantitatively defined. This significantly detracts from many of the technical accomplishments in the manuscript. Specifically, terms such as "endowing", "empowering", "sculpting", "perceive", "revolutionize", "unlocking", "see" and "eyes", "synergized", etc...
3. The presented approach uses light sheet microscopy to view the printed structure. This is an expensive and specialized technique and involves delivering additional light to the sample. This has clear implications on cumulative deliver of light to cells that could affect viability and function. What are the realistic limitations in the fabrication process due to this? Is there an estimate on the total energy delivered to the cells and at which wavelengths? Similarly, light sheet imaging is typically limited to small volumes on the order of a few cubic centimeters are most. So how scalable is the approach really for building therapeutic scale tissues for human use?
4. The authors state that "With GRACE, we demonstrated on-the-fly generation of 3D models to create positive and negative features, including targeted capillary-like vessels that can precisely reach cells, cell clusters and organoids of interest, resulting in improved viability and functionality of the bioprinted cells." However, it is not clear how this has been rigorously demonstrated. For example, what are the positive or negative controls? How is cell viability improved as compared to just redesigning the whole structure and printing it that way?
5. Figure 1 has a lot going on and the caption does not adequately describe it. Both sub-panels 'a' and 'b' have multiple abbreviations that are not defined, optical elements that are not described, and other concepts that may have a graphic but no description in the caption text. Basically, the figure is trying to do too much and as a result does not provide adequate information to the reader.
6. There is a line referring to "digitally encoding a light sheet pattern (i.e. a single column of activated pixels) onto the digital micromirror device (DMD)" as an alternative to the light sheet system described. There appears to be no data referenced for this and not clear why it is included. It should be removed or data added.
7. Results presented in Figure 2 demonstrate that spherical hydrogel structures can be detected and that other hydrogel structures can be printed around them. This is indeed a novel result. However, what is not supported by the data presented is that these are "vascular-like channels". First, they do not appear to be channels and instead appear to be solid hydrogel. If there is any lumen then it is not qualitatively or quantitatively described. Similarly, there is no perfusion through the "channels". The authors also refer to "convoluted capillary balls" around particles. But beyond a general passing resemblance this is not shown. Indeed, capillaries are on the order of 5-10 micron in diameter and these vessels are 2 orders-of-magnitude larger in diameter.
8. In Figure 2 it is also difficult to fully determine what is a computer-generated graphic and what is experimental data from a

microscope. I think the images with black backgrounds are from the microscope and everything else is computer generated, but I am not sure.

9. The example of the femur automatic alignment in Figure 2 is a nice feature. What detracts is that this is a structure that could be bioprinted using other methods. It would be more impactful to show this approach used to print something that other techniques could not achieve.

10. Authors state that “the workflow [is] able to accommodate virtually any array of parametric models and data streams for generating context-driven architectures.” However, it is not clear how the examples given can be extrapolated to support this statement.

11. It is clear how opaque features can be addressed, and the authors do a good job demonstrating this. However, what about air or other gas bubbles that might be in the resin bath? Would these not act as lenses that would refract the light? How would this be addressed? This is a real potential issue when placing already printed objects in the resin bath that might have rough and/or hydrophobic surfaces, or when combining VAM with extrusion printing.

12. The correction for the occluding objects is well described and impressive. What is not clear is what the quality of the correction enables in terms of fabrication. For example, can it enable the formation of a capillary like network similar to the previous Figure 2? A better description of limitations and how this could see some sort of application would support the broader impact of this work.

13. For the bioprinting section, the ability to create a vascular-like structure around the printed cells by combining extrusion and volumetric printing is compelling. However, it is not clear why GRACE is needed, it seems like this structure could easily be predefined in CAD and just printed. A better explanation is needed.

14. In terms of the GRACE versus random channels and insulin secretion, the data shows it is better, but why is GRACE better? That is not clearly explained.

15. It is also unclear how the parametric vessels in Figure 4 are perfused, what flow rates were used, what the wall thickness of the vessels are and the corresponding internal diameter? Also, were these seeded with endothelial cells?

16. The cell densities printed in GRACE look to be in the range of 5-10 million cells/mL. This may be fine for cartilage but is quite low compared to the cell densities found in many tissues, which typically range from 200-600 million cells/mL. It would be quite impactful to see GRACE work with those cell densities.

17. The discussion section should elaborate more on potential applications beyond bioprinting. How could GRACE be adapted for microfluidics, optics, or soft robotics?

18. The manuscript provides impressive visual demonstrations of the GRACE system, but additional quantitative validation would strengthen the claims. It is suggested to add:

- a. Statistical analysis comparing the accuracy of printed structures before and after shadow correction.
- b. Performance benchmarks (e.g., processing time, feature detection accuracy) relative to other existing methods.
- c. A table summarizing success rates for different materials and feature sizes.

19. The mechanical stability of the printed structures is not discussed in detail. Were any mechanical tests performed on the printed constructs?

Referee #2

(Remarks to the Author)

This paper by Florczak et al presents an interesting and well-executed study that demonstrates the potential of a pre-developed workflow integrating imaging and adaptive modelling for volumetric printing. I believe that the study is novel and makes a strong contribution to the field by addressing key limitations of volumetric printing and expanding its functionality. Furthermore, it offers an interesting unique capacity to “adaptation” and “context” into fabrication. The study is of broad interest given the capacity to adapt to multiple printing modalities and that it may be applicable in different areas. In general, the study is timely and novel, and the manuscript is well-written. I believe that the manuscript should be considered for publication once the points below are addressed.

MAIN COMMENTS

While I agree with the authors that the field of additive manufacturing has, for the most part, relied on small progressive increases, I disagree that there has been limited progress incorporating environmental features as part of the fabrication process and into the resulting structures. For example, recent progress incorporating molecular self-assembly (<https://iopscience.iop.org/article/10.1088/1758-5090/ab84cb/meta>) has enabled the printing of structures that both harness the environment and use it to design/fabricate structures with increases structural and compositional complexity. This should be discussed in the introduction.

The method is based on volumetric printing where the volume of ink and internal parts or features remain undistributed by printing. However, the claim that the approach directly applies to other printing techniques seems overstated, without providing supporting examples or a more detailed examination of how it could be implemented. The example given with another technique (extrusion-based printing) seems to generate the file for volumetric printing.

A key limitation of the imaging technique used is the resolution of the particles imaged using it. The diameter of the spheroids used has not been specified but the examples shown suggest that the detection limit is approximately an order of magnitude larger than an individual cell scale. Please elaborate on this to make it easier to understand in the manuscript.

The system offers important advantages (eg speed) and opens new opportunities. However, the setup and development of the system are complex and require the integration of multiple systems. The discussion should present current disadvantages of GRACE and items that could be improved in the future.

Along these lines, it would also be helpful to discuss in more detail how the system can (and can't) deliver "tailor-made tissue mimetic materials". In my opinion, while GRACE offers clear advantages, it falls short from mimicking important features (eg, molecular diversity and compositional complexity) of the inherently complex and dynamic natural environment.

MINOR COMMENTS

In the abstract, please briefly mention what are the "traditional limitations".

P3 Line 3 - "; of which the printer is aware" – repetition?

P3 Line 4 – With regards to the compatibility of GRACE with "a wide range of fabrication technologies", arguably, currently only volumetric printing is shown. Please expand on how it could be adapted to another technique that did not use volumetric printing.

P3 Line 16 – Please specify what you mean by "key tissue components". Do you mean cells or ECM components or both? It is currently confusing as the example provided on vessels is much bigger than the individual cells.

P4. Figure 1a has a lot of information that can be difficult to follow, especially for non-experts. This figure 1 could be divided in two or the figure legend for part a could be expanded as well as the text on page 5.

P6 Line 7 – diameter of aligned particles used 0.3-1.8. The smallest size is ~10X larger than typical cell diameters. It would help to give an example of particles this large.

P6 line 8 and 20 – When describing the size of the spherical alginate particles, these are first described in diameter and then switched to radius. Please use the same.

P9 Line 6 – "especially for VAMS" – only for volumetric printing?

P11 Line 17 – "(iii) the compatibility of GRACE with printing techniques other than volumetric printing". It is unclear if this has been achieved.

P13 Line 9 – Please provide a justification for the use of the surface area as control to allow comparison between the two channel designs.

Unless I missed it, I believe that the "dynamic" conditions have not been specified.

P14 Line 1 – Please provide the diameters of the spheroids used.

Supplementary Information. Please revise as multiple symbols seem to be missing from text and equations.

Fig 4j. Please include scale bar.

Referee #3

(Remarks to the Author)

This manuscript introduces a remarkably capable set of techniques and enhancements of tomographic volumetric additive manufacturing that enable printed structures to be interfaced precisely with dispersed biological cells and other embedded structures, which may occlude or scatter light. The demonstrations are technically impressive and well structured, providing convincing evidence that indeed the GRACE framework can improve reconstructed dose distributions in the presence of, e.g., pillar arrays (Fig 3). Printing fidelity to the intended geometry is competitive with the state-of-the-art and well quantified in the results.

One topic that may bear a little further discussion, perhaps in the supplementary sections, is how the GRACE workflow anticipates and potentially corrects for misalignments between the axis of the material container and the axis of rotation of the printing system (leading, potentially, to 'wobble'-induced misalignments). It seems that GRACE is likely to be inherently able to detect and correct for such misalignments while determining the locations of embedded objects around which vessels or other structures are to be printed. The authors may wish to bear in mind or compare their approach to this very recently published work from the McLeod group which focuses specifically on container misalignment:

<https://doi.org/10.1364/OE.540200>

One minor correction: a number of symbols in the SI appear corrupted in the PDF provided, especially in sections S3, S6 and S7.

Version 1:

Reviewer comments:

Referee #2

(Remarks to the Author)

The authors have carefully addressed my points in the manuscript and done a good job explaining them in the response to reviewers document. I believe that the paper is ready for publication. However, I provide some suggestions below that could strengthen its reach.

I appreciate the authors explanation in the Methods to clarify the reasons behind the detection limit. However, it would help to also include in the Methods (or perhaps in the SI) some of the information presented in the response to reviewers document (answers to my comment 3 and 4). For example, the availability of light sheet imaging systems.

I appreciate and agree with the new text related to my comment #5. To facilitate understanding of non-experts, it would help to provide specific examples or relevant references that can illustrate more tangibly what these "complex multi-component structures" are both in nature and what the authors envision to eventually may be fabricated.

Referee #3

(Remarks to the Author)

Thank you for the comprehensive response to my comments. My comments have been fully addressed in the revised manuscript.

Review of responses to Reviewer 1, Adaptive and Context-Aware Volumetric Printing, by Florczak et al.

Comment 1

The authors provide a suitable response, by toning down the language.

Comment 2

The authors appear to have been responsive, although they don't list the specific word changes they've made in their response. Some of the reviewer's word-choice concerns seem excessively conservative to me.

Comment 3

The authors have provided an extremely detailed response, quantifying the light doses delivered by the light-sheet imaging in the GRACE system and comparing them to literature values and their own experimental observations to confirm that phototoxicity is not a limitation of the approach for the wavelengths and cell types they studied.

As the authors point out, the microscopy scanning mode could be changed from radial to linear to reduce peak imaging dose, or alternative illumination/imaging methods could be used to reduce the overall dose delivered. In any case, I do not think these imaging details, while important, are central to evaluation of the capabilities of the GRACE method.

The response about scalability of light sheet imaging is convincing – to paraphrase, light sheet imaging is needed to develop cm-scale, biologically realistic constructs for research applications, and it may not be necessary to image full organ-sized structures for GRACE to be valuable. I am not sure that I quite follow the suggestion of using 'movable vats' to increase the printable/imageable volume.

Overall, the responses seem satisfactory to me.

Comment 4

The authors respond appropriately by removing their introductory claim of improved viability and clarifying the design of their control experiments.

Comment 5

Splitting the figure into two seems like a good decision and the revised captions appear clear to me.

Comment 6

The figure added is convincing and the explanation of why it may be beneficial to generate the light sheet by DMD is clear.

Comment 7

The response does a good job of elucidating the perfusion of the printed channels. The revised manuscript is clear as to the diameters of channels that have been manufactured with GRACE, but, as far as I can tell, it does not directly address the reviewer's comment that capillaries in actual tissue are 5-10 microns in diameter whereas those demonstrated with GRACE are 10-100 times larger. I suppose there need not be an obligation on the authors to point this out, although the printable size of capillaries is a key current question in bioprinting, so it might be nice to see the outlook for printing smaller capillaries addressed e.g. in the conclusion/future work section.

Comment 8.

The response appears to address the comment satisfactorily by updating the caption.

Comment 9.

The authors have added significant new material on auto-alignment of structures, including a ball and socket joint, so their

modification is highly responsive. I would question, however, the validity of the reviewer's original complaint that the ability to print cartilage onto a femur using other printing methods detracts from the value of that original demonstration: what is new here, surely, is the automated alignment of the printed cartilage to the shape/orientation of the femur in the printing volume, which, to my knowledge, has not been demonstrated with anything close to the same speed or accuracy with other methods.

Comment 10.

The modification to the text made by the authors seems appropriate to address this comment.

Comment 11.

It seems at this point like the reviewer is desperately looking for things to complain about, but the authors provide a fair response, pointing out that compensating for trapped gas bubbles is out of the scope of this manuscript, and that it is easy to avoid incorporating bubbles into the system.

Comment 12.

This response is comprehensive and clear, and it conclusively shows an example of how the occlusion correction algorithm enables a complete print of a trifurcated channel within an occluding geometry that would fail without the correction.

Comment 13.

The response provides a credible explanation as to why a complex design cannot be printed purely by extrusion but requires a combination of extrusion and volumetric light-based printing.

Comment 14.

The text modification satisfactorily addresses the reviewer's question about why GRACE gives superior results.

Comment 15.

This response clearly answers the reviewer's question. I suspect the reviewer would be slightly disappointed with the answer, as clearly uncontrolled, non-unidirectional flow within non-endothelialized channels that are 10-100 times larger than real capillaries leaves quite a lot to be desired in terms of physiological realism, but this manuscript is fundamentally about demonstrating an emerging technology platform and it must surely be accepted that there will be plenty of future room for development of the technique's capabilities.

Comment 16.

The authors provide a convincing response. 5-10 million cells/mL is competitive with state-of-the-art bioprinting techniques, as far as I am aware, and as the authors point out, post-printing proliferation is a credible route to achieving physiological cell densities after using the printing method to template the different cell types within a construct. They also show GRACE printing around spheres of very high cell densities, apparently further showing the value of the occlusion compensation algorithm.

Comment 17.

The authors have graciously modified the discussion in accordance with the reviewer's question.

Comment 18.

The authors comprehensively respond to this demanding question by showing new printing results with two different materials to illustrate dimensional consistency across material types with the method.

Comment 19.

The authors have added compressive testing data to respond to this question.

First of all, we would like to thank the reviewers for their time and for sharing their comments on our manuscript. Based on their feedback, we have now thoroughly revised the document. Our replies are provided in a point-by-point format in this rebuttal letter. Please note that, where needed, we have added numbers to the points raised by the reviewers, to facilitate referencing. Please also note that our answers are marked **in blue**, text taken from the manuscript is marked *in blue and in italic*, and part of the text newly added to the manuscript during the revision is **in blue, italic and highlighted in yellow**. References reported in the rebuttal letter are referred in the bibliography at the end of this letter. High resolution versions of the images are provided attached to the submission, in order to avoid loss of resolution during compression into the PDF file.

Please also note that in order to meet the to keep the main body of the manuscript to maximum 3000 words, we have also shortened the text and moved part of the discussion points and new data to the supplementary files or to the Methods section. When relevant, the new position of the revised text is detailed in this rebuttal letter, to facilitate the task of the reviewers in locating the changes.

POINT-BY-POINT ANSWERS TO THE EDITOR AND REVIEWERS

Referee #1 (Remarks to the Author):

Summary: The manuscript presents an innovative approach to Generative, Adaptive, Context-Aware 3D Printing (GRACE). The integration of volumetric imaging, parametric modeling, and adaptive fabrication is well-explored, demonstrating potential applications in bioprinting and materials science. However, several areas require further clarification, additional validation, and improved organization to enhance readability and scientific rigor.

We thank the reviewer for the assessment of our manuscript, and for the positive and constructive feedback. We have addressed the suggestions, as detailed in the response to the following points.

Major Comments:

1. The goal of the researchers is well stated, to enable the printer to see the structure that is within the photoresin bath and to generate an additional print based on that first structure (printed or placed). However, in general while the data is a clear first step in that direction, the degree of progress appears overstated. The statement at the end of the introduction that “This approach is compatible with a wide range of fabrication technologies and materials, including complex composites, and living cells with bio-friendly hydrogels, unlocking a new generation of applications in tailor-made tissue mimetic materials and data-driven rapid prototyping.” Is overstated, the presented data does not support this.

We thank the reviewer for the comment. We have revised the text to tune the tone of the statements. We also would like to underline that in our manuscript we did demonstrate the potential application of GRACE on two diverse printing techniques, i.e. on volumetric printing and filamented light fabrication. We recognize that the application in EmVP still resulted in generating print files for the volumetric printer, and we agree that GRACE can show the most flexibility when applied with contactless printing techniques, such as volumetric printing.

In view of this, we have now simplified and focused the sentence, which reads:

“This novel workflow allows the printer to automatically produce, within seconds, volumetric, generative designs that adapt and conform to embedded features within the resins, allowing to

explore diverse applications in data-driven additive manufacturing and to process complex materials, including living cells with bio-friendly hydrogels, while also automating overprinting to create complex multi-component structures.”

The compatibility with other printing technologies is kept as future perspective in the Discussion and Outlook section:

*“The workflow’s versatility extends beyond its current implementation, as the concept **could** be directly integrated into other volumetric or layer-wise printing approaches. For instance, it could be incorporated into xolography, which inherently utilizes light sheet optics²⁸; multiphoton printing, acoustic-based printing^{29,30}, or extrusion printing in suspension baths³¹.”*

2. On an overall note, there are multiple occurrences of language used in the manuscript that lacks technical rigor and seeks to imply impact that is not clear or quantitatively defined. This significantly detracts from many of the technical accomplishments in the manuscript. Specifically, terms such as “endowing”, “empowering”, “sculpting”, “perceive”, “revolutionize”, “unlocking”, “see” and “eyes”, “synergized”, etc...

We thank the reviewer for this suggestion. We have now carefully revised the manuscript to adjust the text and to better reflect our accomplishments.

3. The presented approach uses light sheet microscopy to view the printed structure. This is an expensive and specialized technique and involves delivering additional light to the sample. This has clear implications on cumulative deliver of light to cells that could affect viability and function. What are the realistic limitations in the fabrication process due to this? Is there an estimate on the total energy delivered to the cells and at which wavelengths? Similarly, light sheet imaging is typically limited to small volumes on the order of a few cubic centimeters are most. So how scalable is the approach really for building therapeutic scale tissues for human use?

We thank the reviewer for these pertinent observations, and for the opportunity to provide additional information on these topics of phototoxicity, cost and affordability, and scalability.

Light dose and cell viability. First of all, we would like to underline that volumetric printing at 405nm has been reported by our team and several other authors to preserve very high cell viability post-printing, also when using GelMA as printable material¹⁻⁶. While it is important to consider potential risks of phototoxicity from the added illumination steps, light sheet microscopy is well-recognized for its use in live cell imaging, since it illuminates only a thin section (in our case, 50µm in thickness) of the sample over a short amount of time^{7,8}. To improve the discussion and provide more data, we have now expanded the Supplementary information as follows.

-New section: **S19: Light dose distribution during light sheet imaging and cell compatibility**

Our light sheet is equipped with 450nm, 532nm, and 650nm light sources (output: 40-50 mW), all in the visible range to minimize phototoxicity risks. Importantly, for the cell culture experiments involving the insulin release from pancreatic cells, and the osteal and chondral differentiation from stem cells, scanning was performed with the 650nm source, since the cells were stained with Vybrant DiD as indicated in the Methods section. Red light is being regarded generally as safer than shorter wavelengths, with previous reports of toxicity for doses $>200 \text{ J/cm}^2$ ³, far beyond what delivered in our studies. In terms of light doses, we used scans lasting 75 or 150s for all our cell experiments. The

calculated cumulative light dose distribution in every cross section parallel to the base of the vial for all the three lasers are reported in Supplementary Figure 15.

Supplementary Fig. 15.

Estimated light sheet dose within our system as a function of radial position within the printing volume. Dose estimates are performed for a upper-typical scanning duration of 150s, for **a)** the 450nm source, **b)** the 532nm source, and **c)** the 650nm source.

Due to our choice of using a radial light sheet setting, the delivered light dose peaks at the center of the printing vial, which is illuminated for the whole duration of the scan, and rapidly drops along the radius of the vial, where the voxels are illuminated only shortly during a scan as the light sheet sweeps the volume of the vat radially. Based on our model, outside of this 50 μ m-wide region, at 100 μ m away from the center of the vial the dose steeply drops to 14%, to further drop at 2.6% at 500 μ m, to 1.4% at 1mm, to finally reach 0.09% at the inner border of the vial (7.5mm). Assuming that the lasers are operated at full power, and averaging the energy distribution in the beam profile cross-section, we obtain peak values of ~26, 90, and 66 J/cm² for the 450nm, 532nm, and 650nm lasers respectively, over a 75s scan. This value is confined at the center of the vial, and >99.7% (at 450nm), >96.8% (532nm), and >97.8% (650nm) of the circular cross-section of the vial receives a dose below 1 J/cm², in the hundreds of mJ range, typical of conventional light sheet imaging⁴. To further modify the system and enable scanning while delivering lower light doses, it is important to note that the peripheral regions (>2mm away from the center of the vial) in our vat receive doses in the range 20-400mJ/cm², depending on the wavelength, and such doses were sufficient for feature detection, as shown by cellular structures resolved in these regions. In our design, we prioritized mechanical simplicity and ease of implementation, and therefore we opted for a radial light sheet, leveraging the rotating stage already available within the volumetric printer. Changing the scanning procedure, however, GRACE could be performed using a more classical linear light sheet approach, in which every voxel of the samples is illuminated only once during a scan, therefore delivering a low, homogenous light dose across the sample, comparable to that found in the outer regions of the radial light sheet. Finally, the risk of phototoxicity can be further reduced by using pulsed illumination, or other light dose engineering strategies⁵.

For our printing experiments involving insulin secretion and stem cell differentiation, in which the 650nm laser was used for scanning, we did not observe any abnormal behavior, and cells remained functional over 28 days of culture during the bone and cartilage differentiation assay. Nevertheless, to provide additional information, we assessed the effect of light exposure to the shorter wavelength

laser (450nm) on cell viability, even though this specific laser line was not used in our functional cell printing experiments. Although its peak intensity is about 3-fold lower than the one from the 532nm laser, previous studies found that at comparable wavelengths, the maximum tolerable dose is about 5-fold lower in the blue range³, and therefore we could test the cells in the worst-case scenario. MSC-laden GelMA hydrogels were subject to different light sheet scanning regimes, two “normal” illumination settings, used in this study (scanning time of 75s and 150s) and one condition with an extended scanning time (300s), more than what needed to perform GRACE, as control. Cell viability was assessed one day after exposure to detect photodamage, using a LIVE/DEAD viability kit (calcein AM/ethidium homodimer, Thermo Fisher Scientific). Cylindrical samples (6mm in diameter, 2mm height) were imaged with a fluorescence microscope, and the ratio of live and dead cells was counted from 3 randomly selected images per sample (number of samples per group, n=3). Non-imaged, non-printed samples, casted under a 405 nm light (1.4 mW/cm², 3 minutes), as well as printed-only samples were used as controls. All samples displayed viability values >90% the day after exposure, with the few dead cells distributed seemingly randomly throughout the hydrogel samples indicating that our approach does not compromise cell viability (Supplementary Figure 16). ”

Supplementary Fig. 16.

Representative fluorescence microscopy images of the LIVE/DEAD staining for GelMA-encapsulated MSCs exposed to different light sheet scanning regimes. Samples produced via hydrogel casting or volumetric printing, but not imaged under the light sheet were used as control. For all samples, the ratio of living cells was above 90%, with no statistically significant differences detected (n=3, p=0.0931, DF=14). Scalebar is 100µm.

Moreover, the model used for estimating the light dose is now included in the Supplementary Methods as follows:

S18: Idealised light sheet dose estimates over scanning period

A laser power meter (Coherent OP2-VIS, USA) was positioned at the focal point of the light sheet, and the resultant incident power was measured over a finite length L as bounded by the detector diameter ($L=7.9\text{mm}$). The measured powers were $P_{450\text{nm}}=1.26\text{mW}$, $P_{532\text{nm}}=4.6\text{mW}$, and $P_{650\text{nm}}=3.73\text{mW}$. We modelled our light sheet intensity distribution $I(x,y)$ as gaussian (with beam quality factor $M^2 = 1.3$) along the x -axis, and as a top-hat profile of length L along the y -axis due to mean homogenisation by a Powell lens. This can be expressed as:

$$I(x,y) = \begin{cases} I_0 e^{-\frac{2x^2}{\omega_0^2}}, & |y| < \frac{L}{2} \\ 0, & \text{otherwise} \end{cases}$$

where ω_0 is the gaussian waist along x , L is the top-hat height, and I_0 corresponds to the on-axis peak intensity such that the total integrated power equals P . The power can be expressed as

$$P = \iint_{-\infty}^{\infty} I(x,y) dx dy$$

Which factorises into a gaussian integral along x and a top-hat along y

$$\begin{aligned} &= I_0 \int_{-\infty}^{\infty} e^{-\frac{2x^2}{\omega_0^2}} dx \int_{-\frac{L}{2}}^{\frac{L}{2}} dy \\ &= I_0 \sqrt{\frac{\pi}{2}} L \end{aligned}$$

Solving for I_0 yields:

$$I_0 = \frac{P\sqrt{2}}{\omega_0 L \sqrt{\pi}}$$

A script was then written in MATLAB to approximate the dosage over various scanning durations (discretized at 0.36° angular increments). The resultant radial dosage profile is shown in Supplementary Figure 15.

Finally, we would like to underline that, although we demonstrated GRACE with light sheet imaging, we foresee the technology can be applied with other imaging methods, as we proposed in the conclusions section of the manuscript. We have now expanded this discussion point to refer directly to the risks of phototoxicity and to the results of our experiment, linking this point to the supplementary information:

“Furthermore, the adaptability of GRACE allows for exploring alternative imaging modalities, such as optical tomography (including schlieren, diffraction, coherence, or diffuse), or holographic approaches^{32,33,34}, for non-fluorescent imaging, which could provide a broader safe photoexposure window, although we did not see any relevant phototoxic effects within our experiments (Supplementary Method S18-19, Supplementary Fig.15-16).”

Cost of a light sheet imaging set-up. We concur that light sheet microscopy could be a relatively expensive technique. However, in the recent years, several protocols to assemble affordable light sheet set-ups have been described, including recent advanced hardware with a production cost of ~\$10k ⁹, making the system accessible to most users and laboratories. Moreover, our set-up is considerably simplified compared to classic light sheet microscopy, and can be assembled with few thousands Euro. We have now included this information and a relevant reference in the Methods section, “Light sheet imaging module” section:

“Notably, while light sheet imaging has been historically an expensive technology, new reports describe how to build open source, affordable light sheet systems. In our case, our simplified set-up required only the use of a laser source, a beam reducer, a Powell lens for beam shaping, and a cylindrical lens for focusing the light sheet within the volume – all available as off-the-shelf components.”

Furthermore we have expanded the Supplementary information, detailing the cost for assembling a system similar to ours (at a reduced cost using only one type of laser):

SI. Cost-effective design light sheet illumination

A homogenized light sheet source was constructed using readily available off-the-shelf components. The accompanying price list corresponds to a simplified configuration used in our system, operating at 532 nm with a single laser source. Since the Powell lens introduces controlled divergence along the long axis of the light sheet to achieve uniform intensity, an additional cylindrical lens can be used downstream to partially recollimated the beam (not used in our system). Doing so may reduce angular spread and helps maintain a thinner sheet over a longer propagation distance. Alternatively, a cylindrical lens can be used in place of the Powell lens for simpler, lower-cost implementations, at the expense of some uniformity. When paired with a suitable camera, imaging lens, and appropriate emission filters, this configuration enables an effective fluorescence imaging system at a significantly reduced cost compared to conventional setups.

Item	Part Number	Supplier	Qty.	Unit Price (€)	Total (€)
Laser Source					
532nm 50mW Laser	RLDD532-50-3	Roithner GmbH	1	313.00	313.00
Optics					
f =50mm Cylindrical Lens	LJ1695L2-A	Thorlabs	1	151.17	151.17
f =125mm Plano-convex lens	LA1986-A-ML	Thorlabs	1	47.52	47.52
f =-50mm Plano-concave lens	LC1715-A-ML	Thorlabs	1	45.90	45.90
30° Powell Lens	43-473	Edmund Optics	1	242.00	242.00
Optomechanics					
Small V-clamp	VC3C/M	Thorlabs	1	41.04	41.04
Kinematic Mirror Mount	KM100	Thorlabs	2	38.04	76.08
Lens Mount	LMR1/M	Thorlabs	2	14.97	29.94
Cylindrical Lens Mount	CH2B	Thorlabs	1	70.47	70.47
Large V-Clamp	VC3C/M	Thorlabs	1	41.04	41.04

				Total	1058.16
--	--	--	--	--------------	----------------

Scalability of the approach. This is an interesting point of discussion, though we want to underline that for producing a functional construct of high relevance for biomedical research, upscaling to full-size human dimension is not always necessary. In fact, transplantation for regenerative medicine is only one of the possible applications of tissue engineered and bioprinted constructs. When aiming to produce (humanized) *in vitro* models and micro-physiological systems, which are highly sought to reduce animal experimentation and improve drug testing in biomedical research, miniaturized and cm-scale samples are typically ideal, and speed of fabrication is important to increase throughput in the production steps.

We have now expanded the discussion to account for the potential and limitation in terms of size. In our set-up, we imaged samples as thick as 1.5 cm. The current imaging and penetration depth of light sheet is in the range of several cm, indeed, and the technique has been applied for instance to image whole-body adult mice¹⁰. In parallel, another printing technology that uses a light sheet for polymerization (xolography) has demonstrated prints across 3 cm thick square cuvettes, indicating the light sheet could be used successfully at this distance¹¹. Moreover, novel volumetric printing modalities that allow to upscale the print volume have been recently presented, involving continuous printing over a movable vat¹². In mere terms of addressable size, combining light sheet imaging with these moving vat-systems, would be already sufficient to produce some human tissue, and even organ-scale models. For instance, a human pancreas measures on average 75cm³ with an average thickness of 3 cm¹³, a range that can be addressed with a light sheet imaging modality. Therefore we are confident our approach can be scalable, especially as volumetric printing techniques evolve with further research. Moreover, in line of principle, tissue engineering mainly aims at producing tissues or small portions of an organ, which realistically can be sufficient for transplantation.

That being said, it is important to underline that producing functional human tissues is not just a matter of size of the print, and in fact, the goal of engineering fully functional tissues for transplantation remains highly ambitious, with few but notable success stories. We have now enriched the discussion with the following text, added to the discussion section:

“The GRACE workflow, as introduced in this work, also has direct implications for biofabrication, enabling new possibilities in several key areas. These include the creation of biomimetic scaffolds that can adapt to the spatial distribution of cells or organoids, and the fabrication of tissue constructs with highly controlled microenvironments, which are known as potent regulators of cell function, differentiation, and tissue maturation. These systems could already serve as models for biomedical and pharmaceutical research. While light sheet imaging currently enables scanning of multi-cm sized volumes³³, future work would be necessary to produce constructs having (human) full tissue scale sizes. For instance, combining light sheet imaging and GRACE with movable vats for volumetric printing, could expand the imaging and fabrication range, essentially allowing a form of mosaicking the sample volume³⁴. With larger scales of imaging and printing, moreover, new techniques to mitigate scattering will become increasingly needed³⁵⁻³⁷.”

4. The authors state that “With GRACE, we demonstrated on-the-fly generation of 3D models to create positive and negative features, including targeted capillary-like vessels that can precisely reach cells, cell clusters and organoids of interest, resulting in improved viability and functionality of the bioprinted cells.” However, it is not clear how this has been rigorously demonstrated. For example,

what are the positive or negative controls? How is cell viability improved as compared to just predesigning the whole structure and printing it that way?

We thank the reviewer for bringing this to our attention. The sentence quoted by the reviewer is from one of the introductory sentences on page 3, and is actually the only time the word “viability” appears in the manuscript. While viability was not the direct focus of this investigation, we apologize for this mistake and the confusion it can have generated. Instead, the focus should be on improved functionality, which is what our experiments demonstrate.

Our data on cell culture experiments, reported in Figure 4a-d (page 12), focuses on how the optimized design of the construct, obtained with the GRACE approach, can be applied to improve cell functionality, not viability. We have, therefore, now removed the word viability from that sentence in the revised manuscript.

Moreover, viability *in vitro* has been already demonstrated to be very high in most hydrogel-based bioprinting applications, including in volumetric bioprinting of relatively large structures, as shown by several groups including ours¹⁻⁶, and no significant changes in viability across the experimental samples we produced is expected. Conversely, varying the printed architecture is a powerful approach to modulate cell performance². Therefore, investigating and showing the effect of the printed geometry on a key cellular function is more relevant for applications in tissue engineering and tissue modelling. More specifically, our experiments reported in Figure 4 focused on providing a measurement of cell functionality (secretion of the hormone insulin), with hormone synthesis and secretion being a superior function to just viability.

Regarding the controls, the experiment from Figure 4 actually already had a control (a negative one, the “bulk” group). In fact, this was originally described in the Methods section:

“Additionally, two control groups were produced: i) a bulk cylindrical structure containing no channels printed around the extruded toruses as negative control, and ii) a set of randomly generated channels running along the length of the cylindrical bulk, also maintaining the same 180 ± 10 mm² surface area.”

We apologize if that was not clear in our manuscript, and we have now clarified this in the revised manuscript (page 13), making this explicit in the text and in the figure 4 caption.

“To evaluate the efficacy of this approach, we compared these adaptive structures to two controls: randomly generated channels of the same surface area (Fig. 4c), and a bulk structure without channels, as negative control.”

Figure 4 caption

“[...] d) Graph showing the amount of insulin stored and released by β -cells via the bioluminescent reporter nanoLuc for the GRACE print, as well as for the random and bulk controls[...].”

Further, regarding the discussions on cell viability upon light exposure, please refer to our answer to your question number 3.

Finally, concerning the question of the reviewer about comparing GRACE-produced architectures to a pre-designed structure, we understand this is connected to question 13 from the same reviewer. Therefore, we refer the reviewer to our answer to that question.

5. Figure 1 has a lot going on and the caption does not adequately describe it. Both sub-panels ‘a’ and ‘b’ have multiple abbreviations that are not defined, optical elements that are not described, and other concepts that may have a graphic but no description in the caption text. Basically, the figure is trying to do too much and as a result does not provide adequate information to the reader.

We thank the reviewer for this suggestion. As a similar comment was also raised by Reviewer #2, we have now split the figure, into a main figure (what formerly was panel a), and an Extended Data Figure 1 (which comprises what before was panels b and c). To improve clarity, we have revised the captions as well.

Fig. 1 | Experimental GRACE printing. A schematic of the experimental device, with the light sheet (green), imaging, and printing (violet) optical paths indicated. The printing path comprises of a 405nm continuous wave laser source (CW405), collimating lens (L1), fibre coupling lens (L2), square core multimode-fibre optic (FO), collimating lens pair (L3 and L4), DMD, a 1:1 magnification 4f-relay and Fourier filter (L5, Iris, and L6); and folding mirror (MI). The light sheet path comprises three collimated laser sources at 450nm, 532nm, and 650nm (CW450, CW532, and CW650 respectively), combined via dichroic beam combining optics (DM1 and DM2), 2.5x beam reduction optics (L7, L8), a 30° Powell lens (PL), and cylindrical lens (CL) for focusing the light sheet. Along the imaging path, a monochromatic CMOS sensor, with imaging lens (IL) captures fluorescent signals via fluorescence imaging filter (F1). All three optical paths intersect upon the central axis of a resin-filled print vial, coupled to a rotational stage. To minimize refractive errors during both printing and imaging, the vial itself is immersed within a liquid-filled (refractive index matching fluid, depending on the resin to be printed, but typically water when working with low concentration hydrogels) quartz cuvette (CUV) whose optical faces are precisely aligned orthogonal to each path.

Extended Data Fig. 1 | A flowchart illustrating the GRACE workflow. a) A flowchart of the GRACE methodology. Top row: a printable volume, laden with features is presented; the volume is then scanned with light sheet imaging, and, via segmentation and image processing of the polar stack, the volumetric coordinates of the features is obtained in cartesian coordinates. Middle row: the resultant point cloud is then either further processed using clustering detection to index and determine the centroids of each feature or alternatively used as-is – in all cases being sent to the parametric modeller. The parametric modeller uses the data to generate a bespoke model driven by- and adapted to the scanned features. Bottom row: the parametric model is baked, and the resultant file is then processed via the typical tomographic volumetric printing routines to generate a series of back-projections, which are then sent to the printer. The model is printed and washed, thus revealing the GRACE printed structure. If desired the printed structure can then be light sheet imaged again for re-verification purposes. **b)** A subset of various positive and negative parametric geometries shown in (blue), generated using point-like and bulk scanned features (shown in red). Depicted are spherically wrapped channel networks around feature centroids, spherical features interconnected with struts, encapsulation of point-like structures, channel network creation around arbitrarily shaped features, geometric encapsulation of arbitrarily shaped features, sequential generation of parametric geometry in relation to another. This highlights the versatility of parametric modelling in generating a wide range of complex, feature-driven geometries.

6. There is a line referring to “digitally encoding a light sheet pattern (i.e. a single column of activated pixels) onto the digital micromirror device (DMD)” as an alternative to the light sheet system described. There appears to be no data referenced for this and not clear why it is included. It should be removed or data added.

Thanks for this comment. We have now included information and an experimental demonstration of imaging using the DMD to produce the light sheet. It should be noted that using the DMD from the printer could also be a strategy to further reduce the cost of the device, if necessary. This has resulted in the following figure (Supplementary Figure 1) and the related text:

“We tested our imaging and feature detection accuracy onto suspensions of commercially available standardized polyethylene microspheres (either 150 or 500 μ m) both acquiring images using a DMD-generated and a dedicated external light sheet, and benchmarked the results to imaging via conventional microscopy, showing no difference in size detection accuracy (Supplementary Methods S5, Supplementary Fig.1).”

Supplementary figure 1 in the revised manuscript:

Supplementary Fig. 1.

*Validation of using the DMD as a means of light sheet generation for volume scanning, by using polystyrene particles as reference features for benchmarking. **a,b** A 3D representation of a DMD-generated light sheet formed by encoding a single pixel column onto the DMD, and illuminating with the volumetric printing source; and the externally generated light sheet used primarily throughout this work, respectively. **c,d** An identical volume imaged then reconstructed from a scan using both the DMD-based, and external light sheets respectively. The volumes depicted contain the larger 425-500 μ m fluorescent beads. **e,f** Graphs comparing the dimensions of fluorescent beads randomly distributed in gelatine, as measured by brightfield microscopy, 405nm DMD generated light sheet, and the 450nm, 532nm, and 650nm external light sheet sources, for 425-500 μ m specified beads ($n=6$, $F=0.15$ $p=0.95$, $DF=29$), and 125-150 μ m ($n=6$, $F=0.97$, $p=0.44$, $DF=34$) beads, respectively. The graphs indicate no significant difference and comparable dimensional accuracy in all modalities. Moreover, particles smaller than the minimum detectable voxel size of our imaging system could not be detected, as tested with a batch of standard test particles with nominal average diameter=5 μ m.*

7. Results presented in Figure 2 demonstrate that spherical hydrogel structures can be detected and that other hydrogel structures can be printed around them. This is indeed a novel result. However, what is not supported by the data presented is that these are “vascular-like channels”. First, they do not appear to be channels and instead appear to be solid hydrogel. If there is any lumen then it is not qualitatively or quantitatively described. Similarly, there is no perfusion through the “channels”. The authors also refer to “convoluted capillary balls” around particles. But beyond a general passing resemblance this is not shown. Indeed, capillaries are on the order of 5-10 micron in diameter and these vessels are 2 orders-of-magnitude larger in diameter.

We thank the reviewer for sharing their observation. We respectfully disagree regarding the statement that the channels appear “solid hydrogel”. The channels are visualized in black/dark grey in the images in Figure 2a and 2d-e. We understand that, due to the fact that the figure only shows specific cross-sections within a stack, and given the convoluted geometries of the channels which span across all planes, in some cross-sections it may look as if a vessel is not connected or as if there is a dead end. Supplementary Video 2 shows a full stack of the different light sheet cross-sections, revealing the interconnection between the channels.

To improve clarity, we have now also improved:

- the caption of Supplementary video 2 as follows;

“Alginate particles were embedded within GelMA resin, then using GRACE, spherically wrapping channel networks were generated and printed. The movie shows the light sheet plane sweeping across the entirety of the sample, with highly concentric and convoluted channels clearly wrapping around the particles. The scan is performed with the bulk of the gel stained with Cy3.5, and therefore showing fluorescence in a green laser light sheet scan. The particles and the channels, non-fluorescent at that wavelength, appear in black.”

- the caption of Figure 2 as follows:

“[...]the process from acquiring the raw data, to generating the target model (shown rendered), and finally printing and imaging the resultant construct. The light sheet sections at different depths of the printed gel show in greyscale the stained GelMA, and the (cross-sections) of the convoluted channels printed within the gel, while the unstained particles are circled in red to improve visualization and to distinguish them from cross-sections of the channels.”

Moreover, we have now also added new experimental data, showing the printing of convoluted, vascular-like channels and facilitating visualization of the perfusability both through light sheet scans, and perfusing the lumen with a dye. This additional, new experiment was also performed printing channels using two different materials, and producing channels with different sizes, assessing accuracy, to address also the comment 18c from the same reviewer.

Please refer to the new Supplementary Figure 8, which is now cited in the main text, page 6, and the related caption:

“Further proof of perfusability, as well as a demonstration of producing complex vascular-like structures with other hydrogel resins was shown (Supplementary Methods S9, Supplementary Fig.8)”

Supplementary Fig. 8.

A parametric design of a convoluted vessel network containing spherically wrapped channels was produced using GRACE from scanning a particle-laden sample. Thanks to the versatility of the parametric design, multiple STL files were produced adjusting the target vessel diameter (300, 400 or 500 μ m), without changing the overall design, to facilitate comparison across different prints. a) Light sheet 3D reconstruction of the printed channels (displayed in gray), produced via printing in a PEGDA or a GelMA resin. b) Time-lapse micrographs of the channel network being perfused with an alcian blue-stained solution, demonstrating full perfusability of the network. As the appearance of the channels in a 2D image is highly dependent on the angle of rotation of the sample, an overlay of the printed sample and the STL file at the corresponding orientation is also provided, to facilitate visualization. c) Quantification of the feature size (average diameter of the printed vessels). No significant differences were found between PEGDA and GelMA printed vessels at 500 μ m (number of measured vessels $n=12$, $p=0.80$), at 400 μ m ($n=14$, $p=0.06$), and at 300 μ m ($n=12$, $p=0.27$)

Accordingly, we have also expanded the Supplementary Methods section with the following text:

"S9: Perfusability of GRACE constructs

3D models (in STL format) were generated with GRACE. The channel diameters were set to 500, 400 and 300 μm respectively and the exported CAD files for printing were printed using two hydrogels. The first hydrogel consists of 5% (w/v) Poly(ethylene glycol) diacrylate (PEGDA, Mn 700, Sigma-Aldrich), and 0.1% (w/v) lithium phenyl(2,4,6-trimethylbenzoyl)phosphinate (LAP, Tokyo Chemical Industry). Due to the low viscosity of PEGDA, to prevent sedimentation during the tomographic volumetric printing step, the resin was supplemented with 3% (w/v) gelatin (from porcine skin, 175 Bloom, Sigma-Aldrich), allowing for thermal gelation. The second hydrogel consisted of 10% (w/v) GelMA (as described previously) and 0.1% (w/v) LAP. Samples were printed in a tomographic volumetric printer as previously described (PEGDA: light dose of 190 mJ for the 500 μm , 185 mJ for the 400 and 300 μm structure respectively; GelMA: light dose of 150 mJ for all channel sizes). Samples were mixed with 15 $\mu\text{L mL}^{-1}$ Cy3.5-PEG-SH (Mw = 5 kDa) (Biopharma PEG) to increase contrast when imaging with light sheet microscope. Lightsheet imaging was performed using a 532 nm laser, and the lightsheet image stacks were imported into Fiji (ImageJ v1.54p), and the resulting image stacks were pseudo flat field corrected using the ImageJ plugin BioVoxel, and subsequently imported into Matlab (MathWorks 2024a), with which a 3D image was rendered using the Volume Viewer app. Flow visualizing was performed using a bright field stereo microscope (Olympus, SZ61) to record the perfusion of the printed channels with Alcian Blue. The diameter of the vessels was measured from light sheet cross-sections crossing the middle of a channel, analyzing at least twelve randomly selected vessels per sample.”

In terms of quantification of the lumen size, the main text in page 6 has been clarified as following:

“[...]vascular-like channel networks (diameter=450 \pm 20 μm)[...]”

Moreover, supplementary figure S7 shows the lumen diameter of different, simple straight channels, to show the printing accuracy of the printer. More relevant to the point of the reviewer, Supplementary figure S10 (reproduced below) shows the quantification of the average lumen size of a complex network of vessels. We realize that the caption does not fully explain that, and that the rendering is a (negative) image from the light sheet scan to visualize the vessel network, and it is possible that without this information, a reader could think the channels are actually a positive feature (filled), which is not the case. We have now improved the figure caption as follows

Supplementary Fig. 7.

Print tests of parametrically generated, convoluted channel networks with a 0.4 mm diameter, shown at two densities: **a)** low-density configuration with 5 target nodes; and **b)** high-density configuration with 80 target nodes. For both panels, the left image depicts a render of the target design to be

printed, while the right image is a 3D light sheet reconstruction, segmenting out only the internal vessel network, coloured grey to facilitate visualization. Measured channel diameters d and tortuosity τ are shown for each print.

Furthermore, we have also provided a quantification of the lumen diameter in the new experiment reported in Supplementary Figure 8, as described above.

Finally, we agree with the reviewer that the name “capillary balls” can be confusing, as the size of the channels is higher than actual capillaries, as also can be seen by the quantification described above. The size range is more compatible with larger venules and arterioles. To improve clarity, we have now changed the name to “spherically wrapped channel networks”, or simply “spherical channel network”.

8. In Figure 2 it is also difficult to fully determine what is a computer-generated graphic and what is experimental data from a microscope. I think the images with black backgrounds are from the microscope and everything else is computer generated, but I am not sure.

We thank the reviewer for this comment. The reviewer gave the correct interpretation. To improve clarity we have now improved the captions for the figure as follows:

Fig. 2 | GRACE allows printing adaptive and feature-driven prints with complex geometries. A showcase of GRACE printing by generating targeted features around randomly distributed fluorescently stained alginate spheres detected within the resin. **a)** A spherically wrapped channel network architecture is generated around the alginate spheres, showing the process from acquiring the raw data, to generating the target model (shown rendered), and finally printing and imaging the resultant construct. The light sheet sections at different depths of the printed gel show in greyscale the stained GelMA, and the (cross-sections) of the convoluted channels printed within the gel, while the unstained particles are circled in red (or in green) to improve visualization and to distinguish them from cross-sections of the channels. **b)** Interconnection of randomly distributed alginate spheres with printed struts, showing a render of the target geometry (blue) and resultant light sheet reconstruction in 3D after printing. **c)** Encapsulation; with corresponding light sheet 3D reconstruction post-printing indicating presence of spheres. Parametric discrimination of generated features based on **d)** Size or **e)** Spectral emission. Here, different populations receive either a single grazing channel or a more complex spherical channel network. **f)** Automated alignment of cartilage model to femoral head in a two-part sequential print, with computation for the alignment (post-scanning) taking <5 seconds to perform on a common personal computer. Our transformed cartilage model was printed directly onto the femoral head, resulting in a multi-component sequentially printed construct with correct relative positioning of its components.

9. The example of the femur automatic alignment in Figure 2 is a nice feature. What detracts is that this is a structure that could be bioprinted using other methods. It would be more impactful to show this approach used to print something that other techniques could not achieve.

We thank the reviewer for this suggestion. We have now included additional prints of two more complex structures that can act as mechanical joints, including a ball-and-socket joint, and a gyroidal scaffold structure with an hollow core, in which a dumbbell shaped stem is fit and has play for sliding movements. These images are now in Extended Data Fig.2, and are accompanied by the following text:

Extended Data Fig. 2 | Articulating auto-aligned sequential prints with GRACE. a) A gyroid with internal aperture (blue) is precisely aligned to the axis of a randomly oriented stud-like print (pink), thus forming a sliding cylindrical joint along the shaft. Light sheet reconstructions post-printing and washing are shown, highlighting a perspective view (i.) and half-section views through the top and side planes (ii. and iii.), clearly showing the central aperture of the gyroid. Note that as imaging was performed with the construct standing vertically, the gyroid has visibly slid down the shaft, further indicating two distinct components. **b)** An automatically aligned articulating ball and socket joint, with the ball component (pink) printed first, then randomly resuspended. A socket was then automatically aligned to it via GRACE. The light sheet reconstructions (i. and ii.) of the printed show the articulating ball arm in two different positions, indicated by the arrows.

10. Authors state that “the workflow [is] able to accommodate virtually any array of parametric models and data streams for generating context-driven architectures.” However, it is not clear how the examples given can be extrapolated to support this statement.

We thank the reviewer for the comment. In our work, we have provided demonstrations for producing multiple types of context-driven architectures, via parametric modelling. These are for instance the convoluted, and sphere-wrapping channels reported illustrated in Figure 2, for which in Supplementary figures 6 and 7 we have illustrated examples on how the parametric design can be flexibly modified. Furthermore, we have shown how different set of parameters can be leveraged either alone or in combination (vessel size, number of branches, number of particles to be reached by the channels, desired distance of the vessel from a particle, or total surface area of the channels, the latter point in Figure 4), further showing the flexibility of GRACE. Moreover, also in Figure 2, we reported examples on context-driven encapsulation, in which the GRACE printer detect elements to then parametrically design a shell that is printed around them, as well as parametric models that also print connecting beams between the encapsulated particles. These are examples of both positive and negative (hollow) printed design, proving the versatility of our approach.

Our sentence intends to highlight that, for users interested in trying other forms of parametric design, for example, to make prints that combine both channels and encapsulating elements, this would be possible.

We have now rephrased our sentence removing the part indicated by the reviewer, to avoid causing confusion:

“Our experiments demonstrated how GRACE can adapt to various embedded objects and generate precise, functional geometries with minimal user input after the initial parametric model definition. This level of automation and adaptability would be impractical, if not impossible, to achieve through manual positioning and 3D modelling.”

11. It is clear how opaque features can be addressed, and the authors do a good job demonstrating this. However, what about air or other gas bubbles that might be in the resin bath? Would these not act as lenses that would refract the light? How would this be addressed? This is a real potential issue when placing already printed objects in the resin bath that might have rough and/or hydrophobic surfaces, or when combining VAM with extrusion printing.

We thank the reviewer for this observation. We now enriched our text in the Methods section, mentioning this to suggest caution to the reader in avoiding the generation bubbles.

The solution we describe in our manuscript addresses shadowing, but indeed not gas bubbles, as the physics behind potential artefacts in printing due to optical effects of bubbles (i.e. local changes in refractive index, lensing effects), as well as of other effects (i.e. oxygen inhibition of radical polymerization, introduction of hollow defects in the printed object), is completely different than what observed in shadowing due to opaque or occluding features. Addressing these potential effects is outside of the scope of our approach, and in general, bubble generation can be avoided with careful handling (a common guideline for all 3D printing techniques, for which introduction of bubbles would be a potential cause of artefacts). When necessary, and when the materials used allow this, bubbles could also be removed degassing the sample vial before starting the imaging and printing procedure.

In our work, we focused on enabling printing across opaque features, because solving this challenge can permit to print multi-component architectures not readily possible before. All the printed samples demonstrated in our paper have been produced without introducing bubbles into the system, even prints in which highly hydrophobic polymer meshes (i.e. the cage in Figure 3) were introduced in the resin vat.

We have now included the following sentence in the Methods section entitled “Shadow correction of pillar occlusions”:

“It should be noted that the shadow correction algorithm is not designed to prevent or mitigate artefacts caused by (unwillingly) introducing bubbles in the resin vat. As such careful handling and gentle pipetting or pouring of the resin is always recommended when loading the printable materials into the vials.”

12. The correction for the occluding objects is well described and impressive. What is not clear is what the quality of the correction enables in terms of fabrication. For example, can it enable the formation of a capillary like network similar to the previous Figure 2? A better description of limitations and how this could see some sort of application would support the broader impact of this work.

Thanks for this suggestion. We have now included a demonstration of printing a vessel network, containing a 3-branched channel, in presence of a matrix of occluding elements, resembling a stent-like structure. This concept can have application in producing hydrogel structures with perfusable vessels, while also in the presence of reinforcing scaffolds from stiff polymers that provide structural and mechanical support to the otherwise non-load bearing hydrogels.

The results of this new printing experiment is now reported in the new Supplementary Figure 12, which is cited in the main text in the revised manuscript: “*Furthermore, we demonstrated the correct printing of vessel-containing structures also in presence of a shadowing cage (Supplementary Figure 12.)*” The new data shows that the branched structure could be resolved in all the printing sessions, something that was not possible without the shadow correction algorithm. Please refer to the explanation in the caption of Figure 12, highlighting also the consideration and limitations concerning small deviations in printing accuracy, and to the corresponding Supplementary Methods S13. Moreover we have also included Supplementary Video 5, to facilitate the visualization of the open lumen of the channels printed with the shadow correction algorithm.

“S13: Shadow correction for vascular structures printed within an occluding stent-like cage.

To investigate the printability of negative features and hollow channels in the context of shadow correction, the auto-alignment based workflow as described for the ball-in-cage model, was used to volumetrically print around an occlusive SLA printed stent-like structure comprising of 20 cylindrically wrapped 1mm diameter counter-rotating struts (Supplementary Fig. 13a). The target model - a cylindrical construct containing a trifurcated channel of 800µm diameter, was used to demonstrate printing of a convoluted negative feature within the presence of shadowing artifacts. Samples were prepared with 10% GelMA+0.1% LAP, with the addition of Cy5 to facilitate imaging for analysis. The occluding structure was placed within the vial and filled with the resin. The GelMa was thermally gelled in ice water, then the occluding mesh was scanned by using the light sheet as profilometer, as previously described. The resultant sparse surface reconstruction was then used to automatically align a reference 3d model of the occlusive surface. OSMO was used over 20 iterations to optimise the tomographic back projections and thus mitigate shadowing. After printing, samples were washed in warm PBS then imaged under light sheet for verification. Four samples - both uncorrected, and corrected, were analysed by segmenting out the interior channel using thresholding tools in ImageJ. The resultant models of the interior channels were analysed in CloudCompare (www.cloudcompare.org) with two representative samples shown in Supplementary Figure. 12b. . Additionally, the volumes of the segmented channels were also measured in Rhino3D (Supplementary Fig. 12c), as well as a metric for channel completeness, obtained by determining the height ratio of fully formed channels for each trifurcation, and the height of the entire construct (Supplementary Fig. 12d).”

Supplementary Fig. 12.

a) GRACE printing of a three-branched vessel network in presence of a complex pattern of shadowing elements, representing a cylindrical cage with a stent-like structure. Although the mesh here has been used only as occluding phantom, printing such vascular design could be used to pattern perfusable structures within hydrogels, when these are coupled with mechanically strong, reinforcing scaffolds needed for load bearing applications. b) Representative 3D reconstructions of the vessel surface. The shadowing correction pipeline was able to reproduce the design in four out of four printing sessions, even in presence of the occluding mesh. Conversely, uncorrected design always gave broken or incorrect reconstructions, in which the vessel lumen is clogged at different heights (hence, appearing empty in this 3D view), even if the printed parts had good fidelity compared to the STL. Also for the corrected samples, the print fidelity of the channels was compared to the intended design, and the surface mismatch was colour-coded voxel by voxel. The deviation maps and its associated histogram, show that the corrected design displays a peak of deviation of 0.1mm compared to the STL file, with the tail of the histogram reaching 0.3mm. This indicates the correction algorithm allows to reconstruct objects within the light-occluding cage, a feature that would not be possible without GRACE, though the challenge to deliver the correct light dose within the volume can affect print resolution, and this should be kept in mind when designing parts for specific applications. c) quantification of the vessel volume for all prints (boxes :25th and 75th percentile, whiskers 1.5 iqr, t-test, $n=4$, $p=0.00497$, $DF=6$). The lower vessel volume value for the uncorrected sample is a direct consequence of the fact that part of the object could not be resolved, and the vessels were clogged. d) completeness index calculated for each individual vessel branch (3 in each of the 4 samples), obtained by determining the height ratio of fully formed channels for each trifurcation, and the height of the entire construct. This index shows clearly that in no case it was possible to form completely a single vessel with the uncorrected projections, whereas 100% of the vessels were completed and connected when applying the GRACE-enabled shadow correction (boxes :25th and 75th percentile, whiskers 1.5 iqr, t-test, $n=12$, $p<0.0001$, $DF=11$).

Regarding the limitations of the shadow correction algorithm in the Supplementary information (Supplementary Figure 9) we already provided the results of *in silico* experiments describing the extent of correction that is possible, as a function of the number of occlusion and size of occluding features present in each 2D single plane. While it should be noted that the possible extent of correction

and maximum number of occluding elements per plane is also strongly dependent on the type of geometry to be printed (and therefore an universal answer on what structures can be resolved and which ones cannot be printed is not available), this dataset provides a valuable example, as well as proposes a methodology to simulate the outcomes of the experiment and understand beforehand if a given design is printable under certain occluding conditions.

Supplementary Fig. 9. Parametric sweep of randomly distributed circular occlusions within a 2D region spanning quantities between 1 to 30, and diameters between 0.1mm to 1mm. OSMO was used to optimize a cog-like 2D target geometry, with the resultant output quantified by **a,b)** the Jaccard index, and **c,d)** the Bhattacharyya coefficient.

We have now expanded the discussion on the limitations of the algorithm as follows (see page 20 in the revised manuscript):

“It should be noted that the shadow correction algorithm is not designed to prevent or mitigate artefacts caused by (invertedly) introducing bubbles in the resin vat. As such careful handling and gentle pipetting or pouring of the resin is always recommended when loading the printable materials into the vials. Moreover the efficacy of the corrective algorithm lowers proportionally to the number and size of occluding elements present in each plane within the print volume, as evidenced by the analysis reported in Supplementary Fig.9. Since the exact number, shape and size of occluding elements that can be tolerated for printing is not a constant and depends on the architecture to be printed, it is advisable to perform in silico simulations, as reported in our Methods section, ahead printing experiments, to evaluate printability.”

13. For the bioprinting section, the ability to create a vascular-like structure around the printed cells by combining extrusion and volumetric printing is compelling. However, it is not clear why GRACE is needed, it seems like this structure could easily be predefined in CAD and just printed. A better explanation is needed.

We thank the reviewer for highlighting this point, giving us the opportunity to provide a clearer explanation. The reviewer argues that, theoretically, a design like the one generated via GRACE could be made manually, as the CAD design of the cell ring to be extruded is known to the user.

However, while the design of the ring is indeed known, during extrusion printing the ring (or in general, any extruded structure) undergoes deformations and deviations from the ideal print. This happens because of the highly viscoelastic behavior of both the cell ink and of the suspension bath in which the extrusion is performed. This is especially true for inks containing relatively high cell content, including the one tested in our experiments, which had 50 million cells/mL. This occurrence is well-described in bioprinting^{14,15}, and the deformation causes a loss of shape fidelity. Therefore, any new design made to fit the CAD design for the extrusion printer (i.e. the vessels wrapping around the ring) will be misaligned and dimensionally inaccurate, when not directly damaging the integrity of the extruded ring. Moreover, the extruded sample would need to be physically moved to the volumetric printer and manually aligned, introducing a significant additional source of errors, as to date there does not exist any 3D printer that permits to do extrusion-based printing and volumetric additive manufacturing in a single device. This problem is solved by GRACE, as the design for the volumetric printer is adapted to the actual print (and therefore accounting also for artefacts of the extrusion printer) and aligned in a fully automated fashion.

To show this visually, we have now included data in the new Supplementary Figure 19, showing the differences between the actual geometry of extruded elements, and the theoretical output from the CAD design:

Supplementary Fig. 19.

a) Theoretical target extrusion profile when performing extrusion printing in a suspension bath (embedded printing) using a high cell concentration ink (iβ-cells, 5×10^7 cells/mL in 2% w/v alginate), shown as a render of the CAD design, assuming the strut thickness is comparable to the extrusion nozzle diameter. b) Actual printing result typically obtained when extruding the same bioink in the GelMA suspension bath, clearly showing the deviation from the ideal shape, and the low shape fidelity common for such a high cell content ink.

To further expand our discussion, we have now added the following sentences to the revised manuscript, page 22:

“It should be noted that, albeit the CAD design of the extruded cell structures is known, it is still preferred to apply the GRACE algorithm after 3D imaging of the extruded features, and not directly on the geometry from the CAD file. Generating the parametric design after imaging, in fact, ensures that printing artefact, or loss of shape fidelity due to the viscoelastic nature of the cell-based ink are taken into account. Moreover, this also takes advantage of the ability of GRACE to auto-align the volumetric print onto the features of interest, reducing inaccuracies and errors that could be caused by the manual alignment of the print vial.”

14. In terms of the GRACE versus random channels and insulin secretion, the data shows it is better, but why is GRACE better? That is not clearly explained.

Thanks for this comment. We have now extended the explanation to clarify. Please refer to page 13 in the revised manuscript:

“[...]we observed a significant increase in proinsulin secreted in the supernatant of the adaptive, GRACE-printed structures compared to both the random non-targeted and bulk structures (3.2 ± 0.3 , 2.0 ± 0.4 , and $1.4\pm 0.1 \cdot 10^5$ RLU, respectively) (Fig. 4d). This result suggests superior mass transport within the adaptive geometries, likely due to the improved proximity of the cells to the surface of the channels, and therefore shorter diffusion distances, when compared to randomly distributing channels throughout the whole construct, highlighting the advantage of the context-aware approach.

15. It is also unclear how the parametric vessels in Figure 4 are perfused, what flow rates were used, what the wall thickness of the vessels are and the corresponding internal diameter? Also, were these seeded with endothelial cells?

Thanks for noticing the missing information on the methodology. We previously mentioned that we used a shaking platform, but we have now added further information to the Methods section:

“The printed constructs ($n=6$ for each group) with different generated vessel networks were retrieved and cultured overnight in incubator in presence of RPMI 1640 Medium, GlutaMAX™, HEPES (Gibco, Life technologies) supplemented with 10% v/v FBS and 1% penicillin/streptomycin. Dynamic culture was permitted by placing the samples on an orbital shaking platform (95 rpm), to ensure media displacement and flow.”

As the culture was performed with a shaker, flow and nutrient exchange is facilitated, though the exact flow rate inside the channels is not controlled nor necessarily unidirectional.

Regarding the question on the wall thickness, we understand the reviewer is asking about the thickness of the hydrogel layer between the ring of cells and the printed vessels. In fact, the channels are printed as hollow features within a cylindrical, crosslinked hydrogel matrix. As such, the border of the hydrogel ends exactly where a printed vessel starts.

To provide additional information, we have now added the quantification of the average minimum distance between the cell torus and the closest vessel, as well as the average inner diameter of the vessels. Please refer to page 22 in the revised manuscript:

“From this data, the average minimum torus-vessel distance and the average vessel diameter were calculated (Supplementary Fig. 13).

And to the new supplementary figure 13:

Supplementary Fig. 13.

a) Channel areas as determined via manual segmentation for scaffolds containing EmVP printed toruses. The graph indicates both random, and parametric GRACE-printed channels have equal surface areas as anticipated ($n=3$, $p=0.2622$, $DF=4$). b) Measured distance of proximal channels from the torus at horizontal midplane for both the parametric and randomly designed channels ($n=18$, $p=0.016$, $DF=34$). c) Measured diameters of proximal channels around the torus at the horizontal midplane for both the parametric and randomly designed channels ($n=21$, $p=7 \times 10^{-5}$, $DF=40$).

Finally, the channels were not endothelialized, and the experiments were performed as mono-culture of the pancreatic cells, as described in the Methods section.

16. The cell densities printed in GRACE look to be in the range of 5-10 million cells/mL. This may be fine for cartilage but is quite low compared to the cell densities found in many tissues, which typically range from 200-600 million cells/mL. It would be quite impactful to see GRACE work with those cell densities.

Thanks for this observation. We agree that printing at concentrations of cells in the range of 10^8 cells/mL is desirable for many tissues. Nevertheless, printing at high cell density using tomographic volumetric printing is a major standing challenge for this particular technology, due to the scattering profile of dense cell concentrations⁵. Addressing this major challenge of the volumetric printing technique, which is independent of GRACE, is outside the scope of our work.

In addition, while it is true that 10^8 cells/mL is the physiological range in many native tissues (i.e. heart, liver), including this high cell density from the beginning of the print is only one of the possible strategies. Other approaches in tissue engineering also rely on cell proliferation post-printing, and starting from 10 million cells/mL would only require 5 population doublings to reach that range of cell density. For instance, in a recent pre-print (not yet peer reviewed), our group has shown that relatively low concentrations of cells encapsulated in volumetrically printed structures can colonize rapidly most of the volume within the gel, when the printable hydrogels are designed to promote cell migration and proliferation¹⁶.

Alternatively, our group has also shown that in volumetric printing, it is possible to print constructs that have localized regions with very high cell densities^{17,18}. In this same paper, the experiment with the pancreatic cells reported in Figure 4, is performed printing the torus with a density of 50 million cells/mL, for instance. To further expand on this, in the revised manuscript, we have now provided a first, simple proof-of-concept test for GRACE printing in presence of regions containing 200 million cells/mL, in the forms of cells encapsulated in alginate beads, as these are dispersed into the GelMa bioresin. This approach could be useful for printing constructs with locally few regions at physiological-like cell concentration, that could be used as drug testing models. The new supplementary figure is cited at page 19 of the revised manuscript as “This approach could also be performed in presence of alginate capsules, each loaded with a concentration of 200 million cells/mL (Supplementary Figure 18, Supplementary Video 4).

Supplementary Fig. 18.

*Example of a GRACE printed structure by generating targeted features (channels) next to randomly distributed alginate spheres loaded with a concentration of 200 million MSCs/mL. **a,b** Show two micrographs of two microspheres within different regions the same construct. The hydrogel sample was sliced with a razor blade, and imaged with a microscope. The pictures represent and overlay of the bright field, transmitted light image, showing the bulk of the gel and the printed channels running through it and of the fluorescence of the Vybrant DiD-stained cells (here digitally-colored in cyan). Scale bar = 1 mm. A full-thickness scan of the sample is provided in Supplementary Video 4. **c** Close-up fluorescence microscopy image of a cell-laden alginate bead (scale bar = 500 μ m).*

Supplementary Video 4

Series of light sheet cross-sections, showing GRACE-printed channels that were generated in presence of alginate microspheres encapsulating cells at a density of 200 million cells per mL (MSC, DiD stained). The video shows the channels (empty and open) in black, and the cell-laden hydrogel spheres lighting up when hit by the laser of the light sheet, due to the fluorescence emission of the DiD dye.

The related methodology section has also been added to the supplementary information file. Moreover, we would like to remind that, while this proof-of-concept demonstrates the possibility to pattern objects/vessel around multiple, confined regions containing a concentration of 200M cells/mL, printing at this density is still not possible with hydrogels in which this high cell concentration is homogenously distributed. To enrich the discussion on the limitations of GRACE,

we have also included a final paragraph highlighting this limitation and future perspectives, to conclude that section.

S21: Bioprinting perfusable structures embedding high cell density features with GRACE

MSC-laden alginate particles were produced using the set-up previously described (Supplementary Methods S6). Briefly, MSCs (stained with the Vybrant DiD membrane dye to facilitate imaging) were suspended in a 2% w/v alginate solution in PBS, at a density of 200×10^6 cells/mL. The mix was extruded from the inner needle of the coaxial nozzle, under air flow running from the external nozzle, to produce droplets, which were collected in a bath, filled with a 100 mM. The resulting cell-laden microgels were suspended and mixed into a gelatin methacryloyl resin (GelMA, 10% w/v), with 0.1% w/v LAP as a photoinitiator, and loaded into a vial of the tomographic printer. Upon light sheet scanning and using GRACE, channel networks were generated and printed, targeting individual spheres (Supplementary Fig. 18, Supplementary Video 4). Despite the potential of this approach, further improvements in the volumetric printing process would be highly beneficial. For example, in light-based bioprinting, achieving cell densities on the order of 10^8 cells/mL - typical of cell-dense tissues such as the heart or liver - remains a challenge when the suspension is homogeneously distributed within the resin vat, due to the strong light scattering of dense cell suspensions. Developing new strategies to overcome this limitation would significantly enhance the potential of volumetric printing, and by extension, of GRACE, for biomedical research.

17. The discussion section should elaborate more on potential applications beyond bioprinting. How could GRACE be adapted for microfluidics, optics, or soft robotics?

We thank the reviewer for this suggestion. We have taken this opportunity to expand the discussion on potential applications in soft robotics, among the options proposed by the reviewer. Please refer to pages 14 and 15 in the revised manuscript:

“In terms of future directions, the ease of performing overprinting offered by GRACE could be of interest for several innovative applications, for example in soft robotics, for introducing more refined polymeric skins onto previously printed movable parts, or controlling the geometry of hydrogel-based osmotic actuators overprinted across skeletal-like scaffolds²².”

18. The manuscript provides impressive visual demonstrations of the GRACE system, but additional quantitative validation would strengthen the claims. It is suggested to add:

a. Statistical analysis comparing the accuracy of printed structures before and after shadow correction.

We thank the reviewer for this suggestion. The statistical analysis to compare accuracy parameters before and after shadow correction was actually already present in Figure 3. We have now added some additional information to the figure caption:

Fig. 3 | Light sheet mapping of occluding structures and shadow correction. *a*) A flowchart demonstrating the process of scanning, mapping, and correcting for the presence of occlusions. *b*) OSMO-based reconstruction of a cog-like target model without and with shadow correction when influenced by 10 randomly distributed pillar-like occlusions. Jaccard and Bhattacharya Coefficients demonstrate the relative improvements attained by the correction (mean±s.d., t-test, $n=12$, $p<0.001$, $DF=22$). *c*) Rendering showing the SLA printed pillar occlusions, and cog-like target geometry to be volumetrically printed around them. *d*) 3D light sheet reconstruction of corrected and uncorrected prints, in addition to a single optical section of the cross-section of each. *e*) Render showing the ball-in-cage model, with spherical target geometry inside. *f*) Light sheet reconstruction of the resultant prints following the destructive removal of the occluding cage following printing. *g*) Root-mean-square error of the printed part. A lower value indicates less deviation from the target geometry (mean±s.d., t-test, $n=3$, $p=0.0071$, $DF=4$). *h*) Sphericity of each printed sample. A higher value indicates the sample is more spherical (mean±s.d., t-test, $n=3$, $p < 0.0137$, $DF=4$).

Moreover, we have now further strengthened this with additional quantitative data, by adding analysis on the new prints (of vessels) performed with and without the shadow correction algorithms:

b. Performance benchmarks (e.g., processing time, feature detection accuracy) relative to other existing methods.

We thank the reviewer for the suggestion. We would like to underline that, to date, there is no other method to produce adaptive prints in a feature-driven manner. As such comparison of the GRACE workflow with other methods for feature detection and printing is not directly possible.

Nevertheless, we have included additional data to strengthen the manuscript. The processing time for all the printing scenario used in our manuscript was already detailed in Table 1.

To evaluate the feature detection accuracy, we have now performed a dedicated experiment, to reveal the ability of the system to accurately detect features and their size, and comparing this performance to conventional microscopy, as also reported in our answer to your question #6.

“We tested our imaging and feature detection accuracy onto suspensions of commercially available standardized polyethylene microspheres (either 150 or 500 μ m) both acquiring images using a DMD-generated and a dedicated external light sheet, and benchmarked the results to imaging via conventional microscopy, showing no difference in size detection accuracy (Supplementary Methods S5, Supplementary Fig.1).”

Supplementary Methods, S5: Light sheet comparisons and benchmarking:

“In our work we primarily explored the use of externally generated light sheets to perform feature mapping. An alternative, strategy is also to simply leverage the volumetric printing illumination source itself by encoding a single central column of on-state pixels onto the DMD, and projecting that onto the axis of the printing volume (Supplementary Fig. 1a,b) . Though inefficient in terms of power-delivery - as much of the light is lost to the off-state pixels - and with the potential to cause unintended crosslinking of the sample (unless a different wavelength or short exposures times are used), it non-the-less provides a potentially useful alternative source for reconstructing volumes that could be used with GRACE. We validated this by demonstrating the use of the DMD-based light sheet to reconstruct a volume laden with fluorescent beads and benchmarking it by comparing the diameters with the same reconstructed volume generated using the 450nm, 532nm, and 650nm external light sheet sources.

To perform this, green, fluorescent polyethylene beads of diameters 125-150 μ m and 425-500 μ m (UVPMS-BG-1.0 125-150, and UVPMS-BG-1.0 425-500 respectively; Cospheric, USA) were first suspended within a 5% solution of gelatine and distilled water. The two samples, containing either the larger or smaller particles, were imaged with each light sheet source (405nm DMD, and the 450nm, 532nm and 650nm external light sheets), to generate a 3D reconstruction of each sample (Supplementary Fig. 1c,d). Scanning was performed with a 550-600nm bandpass filter for all but the 650nm external light sheet, as at this longer wavelength no fluorescent excitation of the beads was possible. Instead, the back reflected signal of the 650nm source was used. The 3D reconstructions were then generated, and particle diameters were randomly sampled. The results were also compared with diameters obtained via brightfield microscopy (Leica DMI8) (Supplementary Fig. 1e,f).”

c. A table summarizing success rates for different materials and feature sizes.

Thanks for your suggestion. We would like to underline that feature size and print success rate for different materials is not dependent on the GRACE workflow. Rather, printing resolution at different material is strictly dependent on the printing technology used to produce the final part, in our case, on the performance of our tomographic volumetric printer. For instance, using a printer with a DMD having higher resolution than ours will likely produce smaller feature sizes. Success rate is also rather

dependent on the printability of the materials and their reproducibility in terms of batch-to-batch variation, and not on the GRACE workflow. In particular, given a material that crosslinks as intended (for us, always with GelMA and PEGDA, unless the batch is somehow contaminated), the prints are always completed every time. As such an extensive assessment of these elements falls beyond the scope of our manuscript.

Nevertheless, to strengthen our data, we now provide new experimental results on performing GRACE printing using two different materials polyethylene diacrylate (PEGDA) and GelMA, as described in our answer to your question number 7. Rather than making a table, we have included the feature size measurements in a graphical format as a panel in Supplementary Figure 8 (reprinted below). The comparison between the different materials, shows that in both cases the printed channels approximate well the intended vessel size, and box plot shows also the extent of variability, with the boxes indicating the 25th and 75th percentile, and the whiskers the 1.5 iqr.

Supplementary Fig. 8.

A parametric design of a convoluted vessel network containing spherically wrapped channels was produced using GRACE from scanning a particle-laden sample. Thanks to the versatility of the

parametric design, multiple STL files were produced adjusting the target vessel diameter (300, 400 or 500 μm), without changing the overall design, to facilitate comparison across different prints. a) Light sheet 3D reconstruction of the printed channels (displayed in gray), produced via printing in a PEGDA or a GelMA resin. b) Time-lapse micrographs of the channel network being perfused with an alcian blue-stained solution, demonstrating full perfusability of the network. As the appearance of the channels in a 2D image is highly dependent on the angle of rotation of the sample, an overlay of the printed sample and the STL file at the corresponding orientation is also provided, to facilitate visualization. c) Quantification of the feature size (average diameter of the printed vessels). No significant differences were found between PEGDA and GelMA printed vessels at 500 μm (number of measured vessels $n=12$, $p=0.80$), at 400 μm ($n=14$, $p=0.06$), and at 300 μm ($n=12$, $p=0.27$)

19. The mechanical stability of the printed structures is not discussed in detail. Were any mechanical tests performed on the printed constructs?

Thanks for this comment. We have now included in the Supplementary information data on the mechanical properties of printed GelMA hydrogels. Our data and observations are in line with what previously experienced with GelMA hydrogels at 5-10% w/v, which are sufficiently stable for printing and perfusion applications, as well as for the cell culture tests we performed in this study.

Please refer to the new Supplementary Figure 17, p, now cited in the Methods (Resin preparation and printing) section as “The mechanical characterization of the printed hydrogels is reported in Supplementary Figure 17”.

Supplementary Fig. 17.

Mechanical characterization under uniaxial, unconfined compression of GelMA hydrogels produced via volumetric printing and compared to casted controls. Hydrogels formed from precursor solutions at 5 and 10% w/v were produced and tested. These gels, used in this study, could be manipulated during culture, and they were mechanically stable under perfusion tests or cell culture. The compressive modulus of the hydrogels was found to be lower in volumetrically printed samples, compared to casted samples ($n=4$, $***p<0.0001$, $*p=0.0261$, $DF=12$). This result is in line with what previously reported in the literature⁶, as volumetric printing is performed stopping the photoexposure before the polymerization kinetics reaches its plateau, in order to avoid overcuring of small features, and therefore loss of shape fidelity. Higher crosslinking degree, and therefore mechanical properties,

can only be reached via post-curing, when necessary. Partly cured GelMA resins post-volumetric printing have been previously demonstrated biocompatible and non-cytotoxic⁷.

The methodology for performing the mechanical tests is described in a new section in the supplementary information:

“S19. Dynamic Mechanical Analysis of GelMA Hydrogels

GelMA samples for mechanical testing were prepared at the concentrations used in this work: 10% w/v and 5%w/v GelMA supplemented with 0.1% (w/v) LAP. For each formulation, disk-shaped samples were prepared (diameter=6 mm, thickness=2 mm) either by volumetric printing, or via casting in a custom-made teflon mold and exposing the samples for 5 minutes to a 365 nm lamp delivering an intensity of 1 mW cm⁻²). Both casted and printed GelMA disks were kept in PBS in the incubator (37 °C; 5% CO₂) overnight before testing to reach swelling equilibrium. The mechanical properties of the GelMA hydrogels were tested with the Dynamic Mechanical Analyzer (DMA Q800, TA Instruments) via a uniaxial, unconfined compression test. Samples (n = 4) were subjected to a load phase at 0.1 N min⁻¹ up to 0.3 N. The compression modulus was calculated as the slope of the stress/strain curve in the 10–15% strain range. Results are reported in Supplementary Figure 17.”

Referee #2 (Remarks to the Author):

This paper by Florczak et al presents an interesting and well-executed study that demonstrates the potential of a pre-developed workflow integrating imaging and adaptive modelling for volumetric printing. I believe that the study is novel and makes a strong contribution to the field by addressing key limitations of volumetric printing and expanding its functionality. Furthermore, it offers an interesting unique capacity to “adaptation” and “context” into fabrication. The study is of broad interest given the capacity to adapt to multiple printing modalities and that it may be applicable in different areas. In general, the study is timely and novel, and the manuscript is well-written. I believe that the manuscript should be considered for publication once the points below are addressed.

MAIN COMMENTS

1. While I agree with the authors that the field of additive manufacturing has, for the most part, relied on small progressive increases, I disagree that there has been limited progress incorporating environmental features as part of the fabrication process and into the resulting structures. For example, recent progress incorporating molecular self-assembly (<https://iopscience.iop.org/article/10.1088/1758-5090/ab84cb/meta>) has enabled the printing of structures that both harness the environment and use it to design/fabricate structures with increases structural and compositional complexity. This should be discussed in the introduction.

We thank the reviewer for suggesting this interesting reference. The review paper indicated by the referee shows a considerable amount of progresses in the development of self-assembly and stimuli responsive materials and their use for bioprinting and biofabrication applications.

We believe the point made by the referee is not in contrast with our statements in the introduction. In fact, we instead argue that to date, printers are the ones not able to detect stimuli or environmental

features. This significantly limits the designs that they can produce, and therefore the applications that can be targeted.

Rather, since there is no conflict between designing novel printing technologies, and designing novel stimuli-responsive materials, we feel the reference suggested by the reviewer is better fit the future perspective remarks section of our manuscript.

As such, we have modified the text as follows, to improve our discussion. Please refer to page 15 in the revised manuscript:

“These include the creation of biomimetic scaffolds that can adapt to the spatial distribution of cells or organoids, and the fabrication of tissue constructs with highly controlled microenvironments, which are known as potent regulators of cell function, differentiation, and tissue maturation. In parallel, significant progresses have been made in designing self-assembly materials that can be used in bioprinting approaches, which allow to tune the cellular microenvironment at the (sub)cell-level scale³⁸. Converging these classes of materials with GRACE printing, could further permit to better approximate the hierarchical composition of living tissues at the macro-to-micro scale (via printed architecture), down to the micro-to-nano scale (provided by the structured materials). GRACE also opens new avenues as a tool for adaptively modifying the properties of an object at any timepoint after printing. Such modifications could include spatial-selective grafting of chemical compounds and proteins^{35,36}, establishment of stiffness gradients³⁷ or modulation of viscoelasticity³⁸.”

2. The method is based on volumetric printing where the volume of ink and internal parts or features remain undistributed by printing. However, the claim that the approach directly applies to other printing techniques seems overstated, without providing supporting examples or a more detailed examination of how it could be implemented. The example given with another technique (extrusion-based printing) seems to generate the file for volumetric printing.

We thank the reviewer for this comment. We agree the method is primarily applied for volumetric printing, and that in the EmVP experiments the generated STL file are used by the volumetric printer. We also did demonstrate a proof of applicability for a different fabrication technique (FLight).

We have now addressed this point, as discussed in our answer to point 1 from reviewer 1.

3. A key limitation of the imaging technique used is the resolution of the particles imaged using it. The diameter of the spheroids used has not been specified but the examples shown suggest that the detection limit is approximately an order of magnitude larger than an individual cell scale. Please elaborate on this to make it easier to understand in the manuscript.

We thank the reviewer for this comment. Indeed, the imaging resolution of our light sheet imaging system is 14.47µm. As such, the current set-up is suitable to detect cell aggregates, organoids and spheroids. It is important to notice, however, that this is a limitation of our hardware (mainly, the detection optics), but not of the technique, as we purposely selected a low-cost set up to favour accessibility for other researchers. When aiming to improve imaging resolution, there are several light sheet imaging systems that permit to reach higher, and single-cell level resolution, which could be used in a future iteration of our technology.

We have now added more information and these discussion points in the Methods section, page 16 of the revised manuscript:

“Image acquisition for feature scanning/registration was performed using a monochromatic camera (Alvium 1800 U-240m; Allied Vision, USA) in conjunction with an f=50mm C-Mount lens (MVL50M23; Thorlabs, USA). With this hardware we achieved a resolution of 14.47 μ m along the image plane, suitable for capturing cellular aggregates and organoids. Higher resolution hardware would be needed to extend the detection to the single cell-regime, hence the current system is limited to detecting larger particles.”

4. The system offers important advantages (eg speed) and opens new opportunities. However, the setup and development of the system are complex and require the integration of multiple systems. The discussion should present current disadvantages of GRACE and items that could be improved in the future.

We thank the reviewer for this comment. We recognize that integrating multiple systems is needed for our set-up. At the same time, affordable and open source blueprints to build a tomographic volumetric printer and light sheet imaging systems are nowadays available in public repositories (<https://github.com/computed-axial-lithography/OpenCAL>) and in peer reviewed publications⁹, and these efforts towards Open Science will help more researchers.

Regarding the discussion on items that could be improved in the future, we appreciate the suggestion. We have addressed it as detailed in our responses to reviewer 1 (and in our previous response to your question on the imaging resolution). For example, further discussion points include:

On the disadvantage of the cost, and possible improvements, pages 16-17 in the revised manuscript:

“Notably, while light sheet imaging has been historically an expensive technology, new reports describe how to build open source, affordable light sheet systems. In our case, our simplified set-up required only the use of a laser source, a beam reducer, a Powell lens for beam shaping, and a cylindrical lens for focusing the light sheet within the volume – all available as off-the-shelf components.”

On the scalability of the approach, page 15 in the revised manuscript:

“The GRACE workflow, as introduced in this work, also has direct implications for biofabrication, enabling new possibilities in several key areas. These include the creation of biomimetic scaffolds that can adapt to the spatial distribution of cells or organoids, and the fabrication of tissue constructs with highly controlled microenvironments, which are known as potent regulators of cell function, differentiation, and tissue maturation. These systems could have already direct application as models for biomedical and pharmaceutical research. While currently light sheet imaging allows scanning multi-cm sized volumes³³, future work would be necessary to advance this system to produce constructs having (human) full tissue scale sizes. For instance, combining light sheet imaging and GRACE with movable vats for volumetric printing, could expand the imaging and fabrication range, essentially allowing a form of mosaicking the sample volume³⁴. With larger scales of imaging and printing, moreover, new techniques to mitigate scattering will become increasingly needed³⁵⁻³⁷.”

On the imaging resolution, page 16 in the revised manuscript:

“With this hardware we achieved a resolution of 14.47 μ m along the image plane, suitable for capturing cellular aggregates and organoids. Higher resolution hardware would be needed to extend the detection to the single cell-regime, hence the current system is limited to detecting larger particles.”

On the limitation of printing at high cell densities with volumetric printing, Section 19 (supplementary information) in the revised manuscript:

“Despite the potential of this approach, further improvements in the volumetric printing process would be highly beneficial. For example, in light-based bioprinting, achieving cell densities on the order of 10^8 cells/mL - typical of cell-dense tissues such as the heart or liver - remains a challenge when the suspension is homogeneously distributed within the resin vat, due to the strong light scattering of dense cell suspensions. Developing new strategies to overcome this limitation would significantly enhance the potential of volumetric printing, and by extension, of GRACE, for biomedical research.”

On the limitations of the shadow-correction algorithm, page 20 in the revised manuscript:

It should be noted that the shadow correction algorithm is not designed to prevent or mitigate artefacts caused by (invertedly) introducing bubbles in the resin vat. As such careful handling and gentle pipetting or pouring of the resin is always recommended when loading the printable materials into the vials. Moreover, the efficacy of the corrective algorithm lowers proportionally to the number and size of occluding elements present in each plane within the print volume, as evidenced by the analysis reported in Supplementary Fig.9. Since the exact number, shape and size of occluding elements that can be tolerated for printing is not a constant and depends on the architecture to be printed, it is advisable to perform in silico simulations, as reported in our Methods section, ahead printing experiments, to evaluate printability.”

On the possibility to integrate GRACE printing with smart and self-assembly materials, page 15 of the revised manuscript:

In parallel, significant progresses have been made in designing self-assembly materials that can be used in bioprinting approaches, which allow to tune the cellular microenvironment at the (sub)cell-level scale³⁸. Converging these classes of materials with GRACE printing, could further permit to better approximate the hierarchical composition of living tissues at the macro-to-micro scale (via printed architecture), down to the micro-to-nano scale (provided by the structured materials).”

5. Along these lines, it would also be helpful to discuss in more detail how the system can (and can't) delivery “tailor-made tissue mimetic materials”. In my opinion, while GRACE offers clear advantages, it falls short from mimicking important features (eg, molecular diversity and compositional complexity) of the inherently complex and dynamic natural environment.

Thanks for this comment. We agree with the reviewer that it is not an advantage of GRACE to deliver tissue mimetic materials, and we recognize that our statement of enabling “new applications in tailor made tissue mimetic materials” may sound somewhat confusing. However, it was our intention to highlight the value of testing GRACE in combination with different material in future developments. Therefore, we have now removed the statement, and the sentence now reads (page 3 of the revised manuscript):

“[...]allowing to explore diverse applications in data-driven additive manufacturing and to process complex materials, including living cells with bio-friendly hydrogels, while also automating overprinting to create complex multi-component structures””

In addition, we have further clarified the range of materials that can be processed via tomographic volumetric printing, see page 16 in the revised manuscript:

In terms of compatible resins, tomographic volumetric printing is compatible with processing light-crosslinkable materials, whereas other classes of materials can be added as inclusions in the resin vat, for instance via embedded extrusion printing¹⁵ prior to polymerizing the resin.”

Being a printing technology, GRACE offers advantages in designing the cellular environment, in terms of architecture, up to the resolution achievable with this system (for volumetric printing, this is currently in the range between 40-400µm, depending on the specifications of the device). We have also shown that the GRACE technology greatly facilitates multi-material volumetric printing (via auto-alignment of sequential prints), which is an important step in creating anisotropic structures, needed to build tissue-inspired constructs. Nevertheless, the fine composition and molecular diversity (down to the sub-micron scale) of the native ECM cannot be replicated by GRACE alone, and, in fact, that is not the objective of this technology. Other techniques or materials design approach exist for this, e.g. the molecular self-assembly approaches reviewed in Hedegaard et al., 2020¹⁹, as previously indicated by the referee. This discussion point has been addressed expanding the discussion of the manuscript, as indicated in our answer to your point number 1.

“In parallel, significant progresses have been made in designing self-assembly materials that can be used in bioprinting approaches, which allow to tune the cellular microenvironment at the (sub)cell-level scale³⁸. Converging these classes of materials with GRACE printing, could further permit to better approximate the hierarchical composition of living tissues at the macro-to-micro scale (via printed architecture), down to the micro-to-nano scale (provided by the structured materials).”

MINOR COMMENTS

In the abstract, please briefly mention what are the “traditional limitations”.

Thanks for the comment. We have adapted the text as follows:

“[...]traditional additive manufacturing limitations in automating overprinting and adapting the printed designs to the content of the printable material[...]

P3 Line 3 - “, of which the printer is aware” – repetition?

Thanks for noticing this. Following a suggestion from Reviewer 1, that expression was deleted from the manuscript.

P3 Line 4 – With regards to the compatibility of GRACE with “a wide range of fabrication technologies”, arguably, currently only volumetric printing is shown. Please expand on how it could be adapted to another technique that did not use volumetric printing.

Thanks for the comment. This has now been addressed in the revised manuscript. Please refer to our answers to your question number 2 and point number 1 from referee 1.

P3 Line 16 – Please specify what you mean by “key tissue components”. Do you mean cells or ECM

components or both? It is currently confusing as the example provided on vessels is much bigger than the individual cells.

Thanks for the comment. As indicated in the example that directly follows that sentence (blood vessels reaching cells), we mean larger structures. We have rephrased to “*key tissue components, including structures composed of multiple cells*”

P4. Figure 1a has a lot of information that can be difficult to follow, especially for non-experts. This figure 1 could be divided in two or the figure legend for part a could be expanded as well as the text on page 5.

Thanks for your suggestion. We have now divided the figure in 2 parts, please refer also to our answer to question 5 from referee 1.

P6 Line 7 – diameter of aligned particles used 0.3-1.8. The smallest size is ~10X larger than typical cell diameters. It would help to give an example of particles this large.

Thanks for the suggestion. The size range is well compatible with organoids, which can go to a few hundred micrometer to multi-millimeter size. This has now been clarified in the text as follows:

“[...]a size range compatible with that of organoids¹⁸[.]”

P6 line 8 and 20 – When describing the size of the spherical alginate particles, these are first described in diameter and then switched to radius. Please use the same.

Thank you for noticing this inconsistency. This has now been uniformed.

P9 Line 6 – “especially for VAMS” – only for volumetric printing?

Thanks for noticing this mistake, the text has now been changed to “*volumetric printing*”, as per the reviewer’s suggestion.

P11 Line 17 – “iii) the compatibility of GRACE with printing techniques other than volumetric printing”. It is unclear if this has been achieved.

Thanks for the comment. This has now been addressed in the revised manuscript. Please refer to our answers to your question number 2 and point number 1 from referee 1.

P13 Line 9 – Please provide a justification for the use of the surface area as control to allow comparison between the two channel designs.

Thanks for this comment. This has now been included in the Methods section, page 22:

“[...]also maintaining the same 180 ± 10 mm² surface area, to provide a comparable interface for solute exchange from the printed vessels to the hydrogel.”

Unless I missed it, I believe that the “dynamic” conditions have not been specified.

Thanks for noticing the missing information. This has now been added, please refer to our answer to point 15 of reviewer 1. We have used an orbital shaker to facilitate flow and exchange of the media.

P14 Line 1 – Please provide the diameters of the spheroids used.

Thanks. The diameter is $\approx 150\mu\text{m}$. This information has now been added to the Supplementary information section in which the spheroid preparation is described (Section title: *Preparation of MSC spheroids*).

Supplementary Information. Please revise as multiple symbols seem to be missing from text and equations.

Thanks for noticing. We realized that this issue happened when uploading a word document into the manuscript submission system, and once the PDF is automatically generated on the online portal. When submitting the revised manuscript, we will make sure to upload directly a PDF version of the file which displays correctly the symbols.

Fig 4j. Please include scale bar.

Thank you for noticing this missing element. It has now been added.

Referee #3 (Remarks to the Author):

This manuscript introduces a remarkably capable set of techniques and enhancements of tomographic volumetric additive manufacturing that enable printed structures to be interfaced precisely with dispersed biological cells and other embedded structures, which may occlude or scatter light. The demonstrations are technically impressive and well structured, providing convincing evidence that indeed the GRACE framework can improve reconstructed dose distributions in the presence of, e.g., pillar arrays (Fig 3). Printing fidelity to the intended geometry is competitive with the state-of-the-art and well quantified in the results.

We thank the reviewer for the positive assessment of our work and for highlighting its novelty and technical soundness.

One topic that may bear a little further discussion, perhaps in the supplementary sections, is how the

GRACE workflow anticipates and potentially corrects for misalignments between the axis of the material container and the axis of rotation of the printing system (leading, potentially, to ‘wobble’-induced misalignments). It seems that GRACE is likely to be inherently able to detect and correct for such misalignments while determining the locations of embedded objects around which vessels or other structures are to be printed. The authors may wish to bear in mind or compare their approach to this very recently published work from the McLeod group which focuses specifically on container misalignment: <https://doi.org/10.1364/OE.540200>

We thank the reviewer for recommending this interesting topic for discussion, and for pointing us to this recently published paper. Following the suggestion of the reviewer, we have now added a dedicated section to the supplementary file:

S23: Vial alignment and opportunities for further automation.

Reconstructing intricate geometries with volumetric printing requires precise alignment and careful positioning of the printing vial relative to both the axis of the rotating stage, and the optical projection axis⁷. This is also important for the GRACE workflow, as the success of the approach not only relies on the printing fidelity, but also the accuracy with which features can be mapped. Within our custom system, vial alignment and verticality are achieved by mounting the printing vial onto a 4-DoF positioner, comprising a tip/tilt kinematic V-clamp mount (Thorlabs KM100V/M) and an XY stage (Thorlabs DT12XY/M). Iterative alignment is achieved using a MATLAB routine that employs edge detection to provide an indication of the physical pitch/yaw and XY displacement of the vial, as well as the necessary corrections required. This is typically performed at the start of each print.

For future iterations of the GRACE workflow, recent advances in the state-of-the art, such as those outlined by Seymour et al.,⁶ could be implemented within our workflow to perform comprehensive misalignment correction via the software. Specifically, by leveraging the volumetric imaging capabilities of GRACE, we envision inferring all degrees-of-freedom of misalignment within the vial by mapping and characterising features over a full 360° sweep. In practice, this would involve acquiring polar light sheet scans many angles, detecting how fiducials or boundaries shift and rotate, and fitting those shifts to a geometric model that pinpoints net tilt, offset, or rotational errors. Once these misalignment parameters are known, new tomographic image sets could be generated. This holds true with the assumption of accurate alignment between the light sheet to the DMD projection axis - a condition which can be enforced by using a DMD-based light sheet, as described in Supplementary Methods S5. Through this approach, high-fidelity prints at improved reproducibility could be more easily achieved.

One minor correction: a number of symbols in the SI appear corrupted in the PDF provided, especially in sections S3, S6 and S7.

Thanks for noticing. As commented also to referee #2, we now realized that this issue happened when uploading a word document into the manuscript submission system, and once the PDF is automatically generated on the online portal. When submitting the revised manuscript, we will make sure to upload directly a PDF version of the file which displays correctly the symbols.

Rebuttal letter references

1. Bernal, P. N. *et al.* Volumetric Bioprinting of Complex Living-Tissue Constructs within Seconds. *Adv. Mater.* **31**, (2019).
2. Bernal, P. N. *et al.* Volumetric Bioprinting of Organoids and Optically Tuned Hydrogels to Build Liver-Like Metabolic Biofactories. *Adv. Mater.* **34**, 2110054 (2022).
3. Chansoria, P. *et al.* Synergizing Algorithmic Design, Photoclick Chemistry and Multi-Material Volumetric Printing for Accelerating Complex Shape Engineering. *Adv. Sci.* **10**, 2300912 (2023).
4. Rizzo, R., Ruetsche, D., Liu, H. & Zenobi-Wong, M. Optimized Photoclick (Bio)Resins for Fast Volumetric Bioprinting. *Adv. Mater.* (2021) doi:10.1002/adma.202102900.
5. Madrid-Wolff, J., Boniface, A., Loterie, D., Delrot, P. & Moser, C. Controlling Light in Scattering Materials for Volumetric Additive Manufacturing. *Adv. Sci.* (2022) doi:10.1002/advs.202105144.
6. Gehlen, J., Qiu, W., Müller, R. & Qin, X.-H. Tomographic Volumetric Bioprinting of Heterocellular Bone-Like Tissues in Seconds. *SSRN Electron. J.* (2022) doi:10.2139/ssrn.4022139.
7. Hobson, C. M. *et al.* Practical considerations for quantitative light sheet fluorescence microscopy. *Nat. Methods* (2022) doi:10.1038/s41592-022-01632-x.
8. Power, R. M. & Huisken, J. A guide to light-sheet fluorescence microscopy for multiscale imaging. *Nat. Methods* **14**, 360 (2017).
9. Chen, Y. *et al.* Low-cost and scalable projected light-sheet microscopy for the high-resolution imaging of cleared tissue and living samples. *Nat. Biomed. Eng.* **8**, 1109 (2024).
10. Cai, R. *et al.* Panoptic imaging of transparent mice reveals whole-body neuronal projections and skull–meninges connections. *Nat. Neurosci.* **22**, 317 (2019).
11. Regehy, M. *et al.* Xolography for linear volumetric 3D printing. *Nature* **588**, 620 (2020).
12. Boniface, A., Maître, F., Madrid-Wolff, J. & Moser, C. Volumetric helical additive manufacturing. *Light Adv. Manuf.* **4**, 124 (2023).
13. DeSouza, S. V. *et al.* Pancreas volume in health and disease: a systematic review and meta-analysis. *Expert Review of Gastroenterology and Hepatology* (2018) doi:10.1080/17474124.2018.1496015.
14. Schwab, A. *et al.* Printability and Shape Fidelity of Bioinks in 3D Bioprinting. *Chem. Rev.* (2020) doi:10.1021/acs.chemrev.0c00084.
15. Jeon, O. *et al.* Individual cell-only bioink and photocurable supporting medium for 3D printing and generation of engineered tissues with complex geometries. *Mater. Horizons* **8**, 1529 (2019).
16. Falandt, M., Bernal, P. N., Longoni, A., Buchholz, M. & Català, P. Hybrid supramolecular-covalent bioresin promotes cell migration and self-assembly in light-based volumetric bioprinted constructs. *bioRxiv (pre-print)* (2025) doi:10.1101/2025.01.06.631505.
17. Ribezzi, D. *et al.* Shaping Synthetic Multicellular and Complex Multimaterial Tissues via Embedded Extrusion-Volumetric Printing of Microgels. *Adv. Mater.* **35**, 2301673 (2023).
18. Ribezzi, D. *et al.* Multi-material Volumetric Bioprinting and Plug-and-play Suspension Bath

Biofabrication via Bioresin Molecular Weight Tuning and via Multiwavelength Alignment Optics. **2409355**, 1–16 (2025).

19. Hedegaard, C. L. *et al.* Integrating self-assembly and biofabrication for the development of structures with enhanced complexity and hierarchical control. *Biofabrication* **12**, 032002 (2020).

Response to the additional minor comments from Reviewer 2

1) I appreciate the authors explanation in the Methods to clarify the reasons behind the detection limit. However, it would help to also include in the Methods (or perhaps in the SI) some of the information presented in the response to reviewers document (answers to my comment 3 and 4). For example, the availability of light sheet imaging systems.

We thank the reviewer for this suggestion in providing additional information on the availability of light sheet imaging systems. Following the recommendation, we have now added a brief additional text to section S1 of the Supplementary Methods, including a reference that beforehand was only cited in our rebuttal letter. This extra information is in addition to the thorough explanation and components list to build the same light sheet used in our work, which we had already provided together with the previous revision. The text now reads as follows:

“As complementary information to the above-mentioned components that can be used to build the same light sheet imaging system used in this study, we would also like to highlight that additional blueprints for custom-made, affordable light sheet imaging systems are also accessible through more peer reviewed publications¹.”

This includes therefore a new reference in the supplementary file:

1. Chen, Y. *et al.* Low-cost and scalable projected light-sheet microscopy for the high-resolution imaging of cleared tissue and living samples. *Nat. Biomed. Eng.* **8**, 1109 (2024).

2) I appreciate and agree with the new text related to my comment #5. To facilitate understanding of non-experts, it would help to provide specific examples or relevant references that can illustrate more tangibly what these “complex multi-component structures” are both in nature and what the authors envision to eventually may be fabricated.

We thank the reviewer for this comment. Following the suggestion, we have further clarified the statement using specific examples, expliciting naming types of constructs developed in our study. Please refer to page 3 of the revised manuscript:

“It also supports complex materials, including living cells with bio-friendly hydrogels, and automates overprinting to create complex multi-component structures, including models comprising multiple tissue types (i.e. vascularized tissues; bone and cartilage in osteochondral models), and mechanical joints with interlocked movable parts, as notable among a broad array of possible printable designs.”